# Unveiling the *Mycodrosophila projectans* (Diptera, Drosophilidae) species complex: Insights into the evolution of three Neotropical cryptic and syntopic species

Stela Machado[1], Maiara Hartwig Bessa[1], Bruna Nornberg[2], Marco Silva Gottschalk[3], Lizandra Jaqueline Robe[1] *

1 Universidade Federal de Santa Maria (UFSM), Programa de Pós-Graduação em Biodiversidade Animal (PPGBA), Santa Maria, RS, Brazil, 2 Universidade Federal do Rio Grande (FURG), Instituto de Ciências Biológicas (ICB), Rio Grande, RS, Brasil, 3 Departamento de Ecologia, Zoologia e Genética, Universidade Federal de Pelotas (UFPel), Instituto de Biologia, Campus Capão do Leão, Capão do Leão, RS, Brazil

☯ These authors contributed equally to this work.
* lizbiogen@gmail.com

**Data Availability Statement:** Concerning deposition of data in repositories, all new sequences were submitted to GenBank, and

## Abstract

The *Zygothrica* genus group has been shown to be speciose, with a high number of cryptic species. DNA barcoding approaches have been a valuable tool to uncover cryptic diversity in this lineage, as recently suggested for the Neotropical *Mycodrosophila projectans* complex, which seems to comprise at least three different species. The aim of this study was to confirm the subdivision of the *M. projectans* complex while shedding some light on the patterns and processes related to its diversification. In this sense, the use of single and multilocus datasets under phylogenetic, distance, coalescence, and diagnostic nucleotide approaches confirmed the presence of at least three species under the general morphotype previously described as *M. projectans*. Only a few subtle morphological differences were found for the three species in terms of aedeagus morphology and abdominal color patterns. Ecologically, sympatry and syntopy seem to be recurrent for these three cryptic species, which present widely overlapping niches, implying niche conservatism. This morphological and ecological similarity has persisted though cladogenesis within the complex, which dates back to the Miocene, providing an interesting example of morphological conservation despite ancient divergence. These results, in addition to contrasting patterns of past demographic fluctuations, allowed us to hypothesize patterns of allopatric or parapatric diversification with secondary contact in Southern Brazil. Nevertheless, genetic diversity was generally high within species, suggesting that migration may encompass an adaptive response to the restrictions imposed by the ephemerality of resources.

## Introduction

The Neotropical region harbors around 13% of the world's biodiversity [1, 2], encompassing five megadiverse countries [3] and three major biodiversity hotspots [4]. Although the

accession numbers are provided in S2 Table. These sequences will be directly released once they appear on a published document, according to NCBI standards. Other relevant data are within the paper and its Supporting Information files.

**Funding:** This study was supported by Universal-CNPq 14/2013, process numbers 471174/2013-0 and 472973/2013-4. S. M. and M. H. B had a research grant from Coordenação de Aperfeiçoamento de Pessoal de Nível Superior - CAPES (Finance Code 001). L.J.R. is research fellow of CNPq – PQ (# 308371/2018-6). The funders had no role in study design, data collection and analysis, decision to publish, or preparation of the manuscript.

**Competing interests:** The authors have declared that no competing interests exist.

processes responsible for the origin of this diversity have been previously discussed [5–7], it is still estimated that for each known Neotropical species, 10 remain unknown [1]. Molecular taxonomy, and in particular the DNA barcoding approach, has contributed significantly to the task of unravelling these high levels of unknown or cryptic diversity [8]. These tools are especially important when speciation occurs in the absence of morphological divergence, a phenomenon known as cryptic speciation [9–12]. The mechanisms responsible for this pattern are not well understood [13], but it has been suggested that sexual selection acting at the level of non-visual signals [14], recent divergence [15], chance fixation of alleles related to epistatic incompatibilities [16], and allopatric speciation associated with morphological stasis or convergent evolution [17] may help to explain the ubiquity of this phenomenon.

The family Drosophilidae includes more than 4,300 species found in almost all biogeographic regions of the planet [18–20]. Different species are associated with diverse resources, such as fruit, fungi, sap, pollen, and rotten leaves [19, 21]. This ecological diversity is largely due to the saprophagous nature of drosophilids that feed mainly on bacteria and yeast involved in fermentation or on fermentation by-products [18, 19, 22]. Nevertheless, many species are also associated with macroscopic fungi, using them as a resource for feeding (eating spores and hyphae [23]), oviposition and/or breeding sites. This is the case for the *Zygothrica* genus group, which encompasses five mycophagous genera: *Hirtodrosophila*, *Mycodrosophila*, *Paraliodrosophila*, *Paramycodrosophila*, and *Zygothrica*.

*Mycodrosophila* Oldenberg, 1914, comprises around 130 obligatory mycophagous species, which depend on fungi in all stages of their life cycle [24]. This genus seems to have originated in the Old World, later migrating to the New World, where it currently occupies the Nearctic and Neotropical regions [18, 20, 25]. Although only 4% of its diversity has been reported in the Neotropics [20, 24], there is evidence suggesting that the diversity of *Mycodrosophila* in this biogeographic region might be underestimated due to sampling bias [26]. In Brazil, for instance, the number of species described in this genus has nearly doubled in the last two years due to integrated taxonomical efforts, associating traditional morphological techniques with DNA barcoding [25, 27, 28]. The latter approach suggested the existence of entirely cryptic species of *Mycodrosophila*, which cannot be easily differentiated even at the level of male genitalia [28].

*Mycodrosophila projectans* (Sturtevant, 1916) has been considered a widely distributed Neotropical species, with records from Mexico to Brazil [25, 29]. However, when some of these Neotropical populations were evaluated through DNA Barcoding approaches using only nucleotide sequences of the mitochondrial cytochrome oxidase c subunit I (COI) gene, at least three different lineages were uncovered [28]: *M. projectans*, *M. projectans* affinis 1 and *M. projectans* affinis 2. However, these results should be interpreted with caution because the sole use of mitochondrial genes may be misleading due to the presence of nuclear paralogues of mitochondrial genes (numts [30]) or "hitchhiking" with maternally inherited endoparasites [31]. The presence of numts can lead to divergent sequences even within the same genome [30], and endoparasites can modify the matrilineal history of mitochondrial genes both from intraspecific or interspecific populations, causing divergence or homogenization at the level of their mitochondrial nucleotide sequences [31]. The endoparasite *Wolbachia* has already been reported in some groups of mycophagous drosophilids, including one species of *Hirtodrosophila*; therefore, mushrooms may be hotspots for horizontal transfer of this endoparasite, where multiple drosophilid species co-exist [32]. Including nuclear gene sequences or morphological markers, along with mitochondrial genes, may allow insight into these evolutionary relationships.

The aim of the present study was to confirm the subdivision of the *M. projectans* complex while shedding some light on the patterns and processes related to its diversification. We

increased sampling for the mitochondrial COI gene and incorporated sequences of the mito-chondrial cytochrome oxidase c subunit II (COII) gene and of the nuclear hunchback (HB) and alpha methyldopa (AMD) genes. We also performed further morphological and ecological niche modelling evaluations. These analyses confirmed the existence of at least three species within the *M. projectans* complex and indicated some putative autapomorphies related to abdominal color and male genitalia patterns. The morphological and ecological similarity of the three species was notable, especially in the face of divergence dating back to the Miocene.

## Materials and methods

### Sampling

Collections were made at 45 sites distributed along Amazonian, Atlantic Forest, and Pampa Bra-zilian Biomes (S1 Table, Fig 1). The Amazonian biome occupies around 6.7 million $km^2$, and its vegetation includes dry land forests, floodplain forests and *igapós* [33]. The Atlantic Forest includes ca 1.1 million $km^2$ along coastal Brazil, with a large latitudinal range encompassing tropi-cal and subtropical regions [34]. The Pampa biome is restricted to the state of Rio Grande do Sul and occupies an area of 176.5 $km^2$, characterized by grasslands with sparse shrubs and forests within the South Temperate Zone [35]. In these localities, active searches for fruiting bodies of macroscopic fungi were performed in forest fragments during daylight. Adults resting or flying over the fungi were collected with an aspirator [36], and in some cases, fungi were returned to the laboratory and maintained until adults emerged. Flies were stored in absolute ethanol, and an ini-tial identification was performed based on external morphology and male genitalia. Only males were used because morphological differences at the aedeagus level provide more accurate species identification. All material was collected in accordance with Brazilian law, under a scientific col-lection license (SISBIO number 28013–12). As practices did not involve vertebrate species, Brazil-ian law does not require approval from the Ethics Committee on Animal use.

### Amplification, sequencing, and manipulation of molecular markers

Total DNA of each adult male was extracted following the phenol-chloroform protocol described by Sassi et al. [37] or using the NucleoSpin Tissue XS kit (Marchery-Nagel). Ortho-logous sequences of the mitochondrial COI gene were amplified using different combinations of the primers TYJ1460 and C1N2329M (5′ ACTGTAAATATATGATGAGCTCATACA3′), modified from Simon et al. [38], HCO1490 and LCO2198 [39], or COIMYCOF (5′ AYTTT ATTTTYGGRGCHTGR3′) and COIMYCOR (5′ WCCTAATGADCCAAADGTTTCY3′), whereas the mitochondrial COII gene was amplified using primers TL2J3037 and TKN3785 [38], or COII3494M (5′ GGNARVAYDRYDCGRTTRTCDAC3′) and COII3400M (5′ ATYGG NCAYCARTGRTAYTGA3′), modified from Simon et al. [38]. The nuclear HB gene was ampli-fied with primers HB106F and HB903R [40] or HBFMYCOS (5′ CATATACGCAAGCA CAAGAACC3′) and HBRMYCOS (5′ GCTCRGCACTGGCMGCAC3′), whereas the AMD gene was amplified with primers AMDEX4F [41] and AMDBW [42]. The obtained amplicons were purified using a solution of 13% polyethylene glycol (PEG) and 1.6M NaCl. Sanger sequencing was performed by Macrogen (http://www.macrogen.com/eng/).

The resulting electropherograms were assembled using the Staden Package Gap 4 package [43], where each contig was individually checked regarding sequence quality, editing polymor-phic sites according to the IUPAC degeneracy code. Orthologous sequences were aligned with the Clustal W algorithm, as implemented in Mega X [44]. Phase 2.1 [45, 46] was used to deter-mine the gametic phase of the nuclear sequences, as implemented in DNAsp 6 [47]. Mitochon-drial datasets were checked for the presence of stop codons and frameshifts to avoid the presence of numts. The ratio between the number of nonsynonymous substitutions per

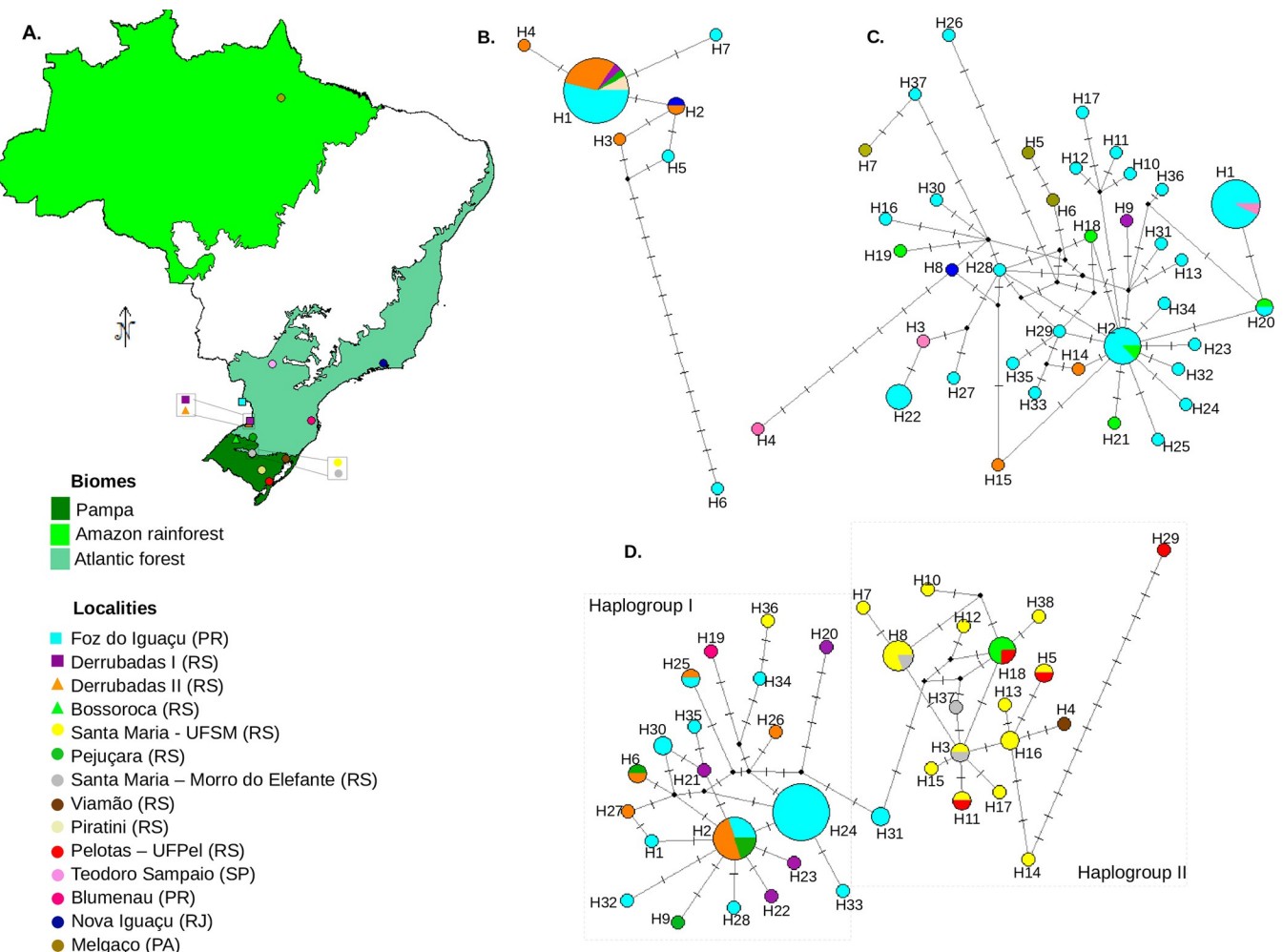

**Fig 1.** Map of sampling points (A) and geographical distribution of COI haplotypes recovered for each species of the *M. projectans* complex: (B) *M. projectans* affinis 1, (C) *M. projectans* affinis 2, and (D) *M. projectans* affinis 3. Brazilian biomes where collections were performed are highlighted on the map, in which squares indicate sampling points where all three species of the *M. projectans* complex were collected in sympatry; triangles indicate sampling points where two species were sampled; and circles represent single species records. Colors represent each different sampling point, according to the legend presented on the lower left margin. On the Networks, each circle represents a different haplotype, whose size is proportional to frequency, and whose colors refer to each locality on the map. Black small circles represent median vectors. Dashes in the lines connecting different haplotypes represent the number of mutations between them. Neotropical and Brazilian Biomes shapefiles were obtained from Löwenberg-Neto [108] and from IBGE [109], respectively.

nonsynonymous site (dN) and the number of synonymous substitutions per synonymous site (dS) was assessed in Mega X using the Kimura 2-Parameters nucleotide substitution model (K2P) [48]. The same method was also used to perform a codon-based Z test of purifying selection (for details, see [28, 49]).

## Morphological and molecular approaches to species delimitation

As Machado et al. [28] employed a DNA barcoding approach suggesting that *M. projectans* was a potential species complex, both morphological and molecular approaches were used here to delimit species. First, adult males were photographed with a stereoscopic microscope to determine abdominal color patterns, and each right wing was removed for further morphometric analysis (see below). Second, using confocal microscopy, we assessed differences in male aedeagus morphology based on Wheeler and Takada [25]. Male terminalia were treated with 10% potassium hydroxide (KOH) and 70% ethanol, following Klaus et al. [50], and

dissected in glycerol. Visualization was performed on a Leica TCS SP8 scanning spectral confocal microscope. Images were acquired using a 20x objective lens, with a 488 nm laser adjusted to full power (11.6 mW) and emission signals detected under 500 to 584 nm.

All individuals were identified by the DNA barcoding in BOLD (Barcode of Life Data) System [51] using COI sequences. Phylogenetic analyses were then performed for each individual gene based on Bayesian inference (BI) to assess phylogenetic structure, as well as reciprocal monophyly of each species. In each case, the best codon partitioning schemes and their respective best substitution models were determined by the Akaike Information Criterion (AIC) [52], as performed in Partition Finder 2.1.1 [53]. BI was performed in MrBayes 3.2 [54], with at least 50,000,000 generations, sampling every 10,000, and burning 25% of the initial trees. Each analysis ended after the average standard deviation between two independent runs was below 0.01, which assures convergence of the results. These analyses were performed using *Mycodrosophila elegans*, MycoNova_SFP_403B1, a sequence characterized by Machado et al. [28] for an undescribed Neotropical species of *Mycodrosophila*, and *D. melanogaster* as outgroups. Sequences for these taxa were downloaded from GenBank (see S2 Table).

Two species delimitation tests based on coalescence approaches that do not require *a priori* species allocation were also employed: the Generalized Mixed Yule Coalescent method (GMYC) [55, 56], which was designed to find genetic clusters evolving independently using single locus datasets, and a multi-locus species delimitation method implemented in the Stacey package [57]. In both cases, substitution models were selected according to the AIC, as performed in jModelTest 2.1.10 [58, 59], employing datasets comprising sequences of the *M. projectans* complex, *M. elegans* and MycoNova_SFP_403B1 (S2 Table). In the first case, ultrametric trees were reconstructed in Beast 2.6 [60], with MCMC encompassing at least 50,000,000 iterations with sampling every 1,000–5,000 chains and burning the first 10%. Further evaluations of the trees were performed using a single threshold analysis in the Splits package (https://github.com/tfujisawa/splits_tmp) in R 1.3.1093 [61], which detects transitions between inter- and intra-specific processes.

Stacey analysis was performed in Beast 2.4.8 [62] under unlinked substitution models, unlinked relaxed molecular clock models, and unlinked Stacey coalescent parameters, with ploidy levels established according to the respective marker. The MCMC encompassed 200,000,000 generations, saving every 10,000, and burning the first 10%. Convergence was evaluated in Tracer 1.60 [63], checking likelihood and posterior probability traces and confirming that all ESS values were above 200. Clustering schemes and species assignments were later evaluated using Species Delimitator Analyzer [64], with a collapse height value of 0.0001.

Additionally, the Automatic Barcode Gap Discovery (ABGD) algorithm was applied for each COI, COII, COI+COII, HB, and AMD dataset, under the Kimura 2-Parameters substitution model, as performed on the website https://bioinfo.mnhn.fr/abi/public/abgd/abgdweb.html. The ABGD groups individuals based on the pairwise distance observed between sequences, without an *a priori* hypothesis [65]. Finally, the assignments obtained by the strategies described above were employed to measure intra- and interspecific nucleotide distances and to obtain diagnostic characters for each species of the individual datasets (COI, COII, COI+COII, HB and AMD datasets). Pairwise distances were calculated on MEGA X using the Kimura 2-parameter nucleotide substitution model. Diagnostic nucleotides were recovered for each putative species using the Spider package [66] in R.

## Morphometric analysis

The right wings of males of the *M. projectans* complex were photographed with a SteREO DiscoveryV20 microscope with the program AxioVs40 V 4.8.1.0 (Carl Zeiss Imaging Solutions

GmbH), using an increase of 37.5X. TPSdig2 [67] was used to mark ten landmarks chosen based on wing venation, representing bifurcations, intersections, and tip veins (S1 Fig, S3 Table). Partial and relative warps, centroid size, and a weight matrix were then calculated with TPSRelwn [68], where a graphic representation from shape variation was obtained. Analysis of variance (ANOVA) was performed to compare the centroid size of the wings within and among species, and a canonical variate analysis (CVA) was performed to compare their relative warps. ANOVA and CVA were performed in PAST 3 [69].

## Diversity and population structure analyses

Diversity estimates (number of haplotypes (H) or alleles (A), haplotype diversity (Hd) or expected Heterozygosity (He), nucleotide diversity ($\pi$), and number of polymorphic sites (*s*)) and neutrality tests (Tajima's D [70], Fu's and Li's D and Fu's and Li's F [71]) were performed in DNAsp 5.10.01 [72]. Genealogical relationships between haplotypes or alleles (COI, COI +COII, AMD, and HB) were estimated by median-joining, as implemented in Network 10.1.0 [73].

Pairwise fixation indexes (Fst) among species and populations were estimated for each dataset in Arlequin 3.5 [74], with 1,000 permutations. Analysis of Molecular Variance (AMOVA) was also performed with the same software subdividing sequences by species (as identified by DNA Barcoding) and populations (according to sampling localities) to assess the covariance among groups or species ($\Phi$ct), among populations within groups or species ($\Phi$sc) and within populations ($\Phi$st). A Mantel test was performed in Arlequin 3.5 to assess the correlation between geographical distances and genetic divergences. Finally, a Bayesian analysis of population structure (BAPS) was performed for mitochondrial markers using BAPS 6.0 [75] to investigate population subdivision without *a priori* assignments. The latter two analyses were performed only for COI, COII, and COI+COII datasets.

## Dating

A chronophylogenetic analysis was performed in Beast 2.6.3 [62] under a multi-species coalescent and a relaxed molecular clock, using the StarBeast package to construct a putative species tree based on the gene trees estimated for COI, COII, HB, and AMD. For this task, in addition to the new sequences characterized here for the *M. projectans* complex, additional sequences were downloaded from GenBank for *D. melanogaster* (*Sophophora* subgenus), *D. virilis* (*Siphlodora* subgenus), *D. ornatifrons* (*Drosophila* subgenus), *D. grimshawii*, and *Scaptomyza* (Hawaiian drosophilids) from the Drosophilinae subfamily, and *P. variegata* from the Steganinae subfamily (see S2 Table). These species represent a set of different lineages within the Drosophilidae [76] and were employed as successive outgroups, as well as to establish calibration points within our phylogeny (see below). Unlinked molecular substitution models were employed for each gene according to the results provided by the AIC performed in jModelTest. Unlinked substitution rates were estimated for each gene based on calibrations to the split between *Scaptomyza* and *D. grimshawii* using a prior with a mean of 17.5 Mya and stdev of 1.25 Mya, according to the fossil data [77, 78] and to the divergence between *Sophophora* and the remaining *Drosophila* with a prior with a mean of 55 Mya and stdev of 8 Mya, according to the average calculated between the molecular clock dating estimated by Tamura et al. [79] (~62.9 Mya) and Suvorov et al. [80] (~47 Mya)]. Unlinked gene trees were also evaluated under a Yule model, with ploidy levels established according to the inheritance pattern of each marker. The analysis was performed for 800,000,000 iterations, storing every 10,000 and discarding the first 10% as burn-in. The run was then visualized in Tracer 1.6 to ensure

convergence and adequate sampling. Results were summarized in TreeAnnotator 1.8.0 [81] and visualized in FigTree 1.4.3 [82].

Extended Bayesian skyline (EBPS) [62] analyses were also performed for each species in Beast 2.6.3, employing COI and COII datasets. This analysis was performed using unlinked substitution models, as selected by the AIC performed in jModelTest. Priors for the unlinked relaxed molecular clock models were established based on the mean of the ucld.means obtained for each gene by the chronophylogenetic analysis described above. Unlinked extended Bayesian models were finally applied for each marker, with population models assigned according to differences in effective population sizes. Each analysis encompassed at least 500,000,000 iterations, sampling every 10,000, and burning the first 10%. These results were evaluated in Tracer 1.6, and skyline plots were constructed in R.

## Species distribution modelling

Potential distribution maps were constructed for each species of the *M. projectans* complex based on environmental niche modelling (ENM) [83]. Bioclimatic variables were first downloaded from the Paleoclim database [84] at a resolution of 2.5 mins, and then cut with a shapefile encompassing the Neotropics. Pearson correlations between the cut layers were estimated in R using the packages Raster [85] and corrplot [86], and the seven least correlated abiotic variables were employed in the modelling strategy: temperature seasonality (Bio-4), mean temperature of the warmest quarter (Bio-10), mean temperature of the coldest quarter (Bio-11), precipitation of the wettest quarter (Bio-16), precipitation of the driest quarter (Bio-17), precipitation of the warmest quarter (Bio-18), and precipitation of the coldest quarter (Bio-19). As cryptic diversity in the *M. projectans* species complex was previously neglected, only the sampling points recorded in this study (S1 Table) were used. To decrease the effect of sampling bias, background points were selected along a radius of 500 km from each sampling point.

Models were generated in R 1.3.1093 using the maximum entropy algorithm function contained in the "dismo" package [87], with calibration to the present and projection to four moments of the past: Late Pliocene (3.3 Mya), Marine Isotope Stage 19 (MIS19) in Pleistocene (MIS, *c.* 787 Kya), Last Interglacial (LIG, *c.* 130 kya), and Last Glacial Maximum (LGM, *c.* 21 kya). In each case, 75% and 25% of the registers were employed as training and test, respectively, in 25 cross-validation replicates. The accuracy of the predictive distribution models was assessed through the Area Under the Receiving Operating Curve (AUC) mean value.

Measurements of abiotic niche overlap between species and identity tests were performed in R using the ENMTools package [88]. For this task, pairwise niche overlap was evaluated according to Schoener's D and Hellinger's I, for which 0 means no overlap and 1 indicates that niches are identical. To identify if observed measures were significantly different from the expectations drawn from the null hypothesis of niche equivalency, the null distribution was derived from 25 replicates, in each of which registers were pooled and randomized and the potential distribution maps were derived under the maximum entropy algorithm.

## Results

A total of 172 individuals first recognized as *M. projectans* were collected, and the following numbers of sequences were characterized: 172 for COI (~727 bp), 70 for COII (~560 bp), 20 for AMD (~921 bp), and 25 for HB (~398 bp) (Table 1; for accession numbers, see S2 Table). Sampling disparity occurred due to frequent amplification failures, which commonly led to the premature termination of some DNA samples. No sequences contained stop codons or frameshifts, and all mitochondrial sequences that did not reject the null hypothesis of neutrality in the codon-based Z test of purifying selection, presenting a dn/ds ratio near one, showed

low K2P distances (d < 1.3%), suggesting stochastic deviation in intraspecific variation (S4 Table).

## How many M. projectans species are there?

**Molecular differentiation.** Of the 172 COI sequences characterized for the *M. projectans* complex, 33 were identified in BOLD as *M. projectans* affinis 1, 61 as *M. projectans* affinis 2, and 78 as *M. projectans*, hereafter referred to as *M. projectans affinis* 3. All individuals were recovered as a highly supported monophyletic lineage in BI performed with COI (PP = 0.93), COII (PP = 1.00), and AMD (PP = 1.00), whereas for HB, individuals of *M. projectans* affinis 1 were recovered as a basal grade (PP ≤ 0.68) (S2 Fig). Except for this grade recovered for *M. projectans* affinis 1 with HB, the three species were reciprocally monophyletic (PP ≥ 0.99). The analyses performed with GMYC also resulted in three clusters for COI, although COII and HB further subdivided *M. projectans* affinis 3 into three and two clusters, respectively, and AMD joined *M. projectans* affinis 1 and *M. projectans* affinis 3 in one cluster (S5 Table). Nevertheless, the delimitation with higher PP recovered in the multi-locus coalescent Bayesian-based analysis executed in Stacey supported the subdivision of the *M. projectans* complex in three clusters (PP = 0.015) (S5 Table), in accordance with the separation seen in the phylogenetic analyses. Finally, the ABGD analysis also resulted in three clusters for COI, COII, COI+COII, and HB datasets, although AMD defined only two groups of sequences, one of which grouped sequences of *M. projectans* affinis 1 and *M. projectans* affinis 3 (S5 Table).

Considering the delimitation in three species, *M. projectans* affinis 1, *M. projectans* affinis 2 and *M. projectans* affinis 3 showed maximum COI intraspecific distances of 4.8, 3.1 and 6.1%, respectively (Table 1). The minimum interspecific distance for this marker was 12.1%, as recovered in a comparison involving individuals of *M. projectans* affinis 1 and *M. projectans*

**Table 1. Summary information regarding sample size (N), divergence (MI%, MX% and MD%) and diversity analyses (π%, H or A, Hd or He and S) and results of neutrality tests (Tajima's D, Fu and Li's D and Fu and Li's F) performed for each of the three species of the *M. projectans* complex with each of the employed mitochondrial and nuclear markers.**

| | | Species | N | MI% | MX% | MD% (SD%) | π%(SD%) | H or A | Hd or He | S | Tajima's D | Fu and Li's D | Fu and Li's F |
|---|---|---|---|---|---|---|---|---|---|---|---|---|---|
| mtDNA | COI | *M. projectans* aff. 1 | 33 | 12.1 | 4.8 | 0.4 (0.1) | 0.3 (0.2) | 7 | 0.383 | 20 | ***-2.32*** | ***-3.52*** | ***-3.69*** |
| | | *M. projectans* aff. 2 | 61 | 12.3 | 3.1 | 0.8 (0.1) | 0.7 (0.1) | 37 | 0.931 | 51 | ***-2.04*** | ***-3.12*** | ***-3.24*** |
| | | *M. projectans* aff. 3 | 78 | 12.1 | 6.1 | 1.5 (0.3) | 1.3 (0.1) | 38 | 0.932 | 49 | -1.29 | ***-3.63*** | ***-3.25*** |
| | COII | *M. projectans* aff. 1 | 9 | 9.7 | 1.8 | 0.4 (0.5) | 0.4 (0.2) | 3 | 0.417 | 10 | **-1.84** | **-2.09** | **-2.27** |
| | | *M. projectans* aff. 2 | 22 | 9.7 | 2.4 | 0.8 (0.5) | 0.8 (0.1) | 14 | 0.896 | 22 | -1.05 | -1.19 | -1.34 |
| | | *M. projectans* aff. 3 | 39 | 11.4 | 2.7 | 1.4 (0.7) | 1.2 (0.1) | 27 | 0.974 | 34 | -0.75 | -2.05 | -1.90 |
| | COI + COII | *M. projectans* aff. 1 | 9 | 12.5 | 1 | 0.4 (0.1) | 0.3 (0.1) | 5 | 0.722 | 16 | ***-1.92*** | ***-2.18*** | ***-2.37*** |
| | | *M. projectans* aff. 2 | 18 | 12.5 | 1.7 | 0.8 (0.1) | 0.7 (0.08) | 14 | 0.954 | 38 | -0.97 | -1.33 | -1.42 |
| | | *M. projectans* aff. 3 | 39 | 12.5 | 2.8 | 1.4 (0.2) | 1.4 (0.05) | 30 | 0.978 | 72 | -0.43 | -1.82 | -1.58 |
| nDNA | AMD | *M. projectans* aff. 1 | 3* | 4.2 | 0.3 | 0.2 (0.1) | 0.2 (0.04) | 3 | 0.8 | 4 | 1.18 | 1.47 | 1.49 |
| | | *M. projectans* aff. 2 | 9* | 4.2 | 2.3 | 1.2 (0.6) | 1.2 (0.1) | 9 | 0.941 | 34 | 0.34 | *1.66* | 1.48 |
| | | *M. projectans* aff. 3 | 8* | 4.3 | 7 | 2.9 (2.5) | 2.8 (0.7) | 12 | 0.967 | 76 | -1.46 | -0.43 | -0.73 |
| | HB | *M. projectans* aff. 1 | 6* | 3.5 | 1.3 | 0.4 (0.4) | 0.4 (0.1) | 5 | 0.667 | 4 | -1.02 | -0.46 | -0.68 |
| | | *M. projectans* aff. 2 | 5* | 2.5 | 2.2 | 0.9 (0.5) | 0.9 (0.2) | 7 | 0.911 | 7 | -0.77 | -0.13 | -0.32 |
| | | *M. projectans* aff. 3 | 14* | 2.5 | 2.6 | 0.6 (0.5) | 0.6 (0.1) | 10 | 0.704 | 13 | **-2.01** | **-2.82** | **-3.01** |

N = number of specimens, MI% minimum interspecific distance in percentage, MX% maximum intraspecific distance in percentage, MD% mean intraspecific distance in percentage with standard error, π nucleotide diversity, H number of haplotypes (for mitochondrial markers), A number of alleles (for nuclear markers), Hd haplotype diversity (for mitochondrial markers), He expected heterozygosity (for nuclear markers), S variable sites. In bold significance of p < 0.05 and in bold and italic p < 0.02.
*The number of alleles analyzed for nuclear markers was doubled after PHASE analysis.

affinis 3. The analysis of diagnostic characters performed for COI resulted in 18 diagnostic sites for *M. projectans* affinis 1, 17 for *M. projectans* affinis 2 and 11 for *M. projectans* affinis 3 (S6 Table). Generally, this pattern was repeated for COII, which showed a barcoding gap of 7.3% for the complex (maximum intraspecific distances of 2.7% and minimum interspecific distances of 10%) (Table 1) and recovered at least 21 diagnostic sites when the subdivision in three species was considered (S6 Table).

**Morphological differentiation.** Individuals of the *M. projectans* complex were extremely similar in external morphology, but three different abdominal color patterns could be identified. Two patterns were shared between *M. projectans* affinis 1 (n = 12) and *M. projectans* affinis 3 (n = 30), where the yellow spots of tergite IV were totally or partially separated by a posterior projection of the black area, hereafter called abdominal pattern A (Fig 2A and 2B; S7 Table). The other pattern, hereafter called abdominal pattern B, was exclusive to *M. projectans* affinis 2 (n = 24) and showed a black background and two yellow spots fully confluent in the dorsal region of tergite IV (Fig 2C; S7 Table). There were two exceptions to this pattern found in this species. A thorough examination of the description of *M. projectans* provided by Sturtevant [88], the redescription of Wheeler and Takada [25], and the holotype photos provided by the National Museum of Natural History (Washington D.C., USA) confirmed the close resemblance of *M. projectans* affinis 2 with the described species.

Wing morphometry was used as a potential morphological marker in species identification for 57 individuals, of which 13 belonged to *M. projectans* affinis 1, 19 to *M. projectans* affinis 2 and 23 to *M. projectans affinis 3*. The ANOVA and CVA tests did not show any statistically significant differences among species regarding the centroid size of wings and the scores of the relative warps, respectively (S3 Fig; S8 Table). In fact, a wide overlap of wing shape was detected among species (S3 Fig), so wing morphology did not provide enough resolution to distinguish species within the *M. projectans* complex.

At the male genitalia level, 2, 9, and 6 individuals of *M. projectans* affinis 1, *M. projectans* affinis 2 and *M. projectans* affinis 3, respectively, were examined by optical microscopy, and 7, 14, and 5, respectively, were examined by confocal microscopy. These evaluations revealed the presence of two male genitalia patterns within the *M. projectans* complex (Fig 3; S7 Table): one type of aedeagus structure was shared between *M. projectans* affinis 2 and *M. projectans* affinis 3 and resembled the illustrations provided by Wheeler and Takada ([25], pg. 396, Figs 15–18) for *M. projectans* (hereafter called type I) (Fig 3A); the second type was exclusive to *M.*

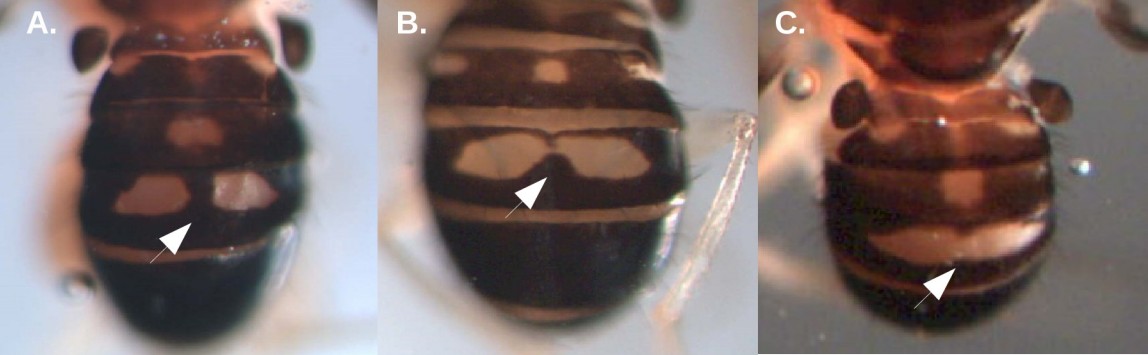

**Fig 2. Abdominal coloration (dorsal view) of the different species of the *M. projectans* complex, as visualized in a stereomicroscope.** (A) Pattern A1, presenting two individual light spots in tergite IV; (B) Pattern A2, presenting two confluent light spots, that are narrowed in the median region, (C) Pattern B, presenting confluent light spots, without the narrowing in the median region. Patterns A1 and A2 are seen in *M. projectans* affinis 1 and *M. projectans* affinis 3, whereas pattern B is exclusive of *M. projectans* affinis 2. Image (A) of sample 577A109, (B) of sample 581A26 and (C) of sample 577A88 (see S7 Table for more details).

## A. Type I

## B. Type II

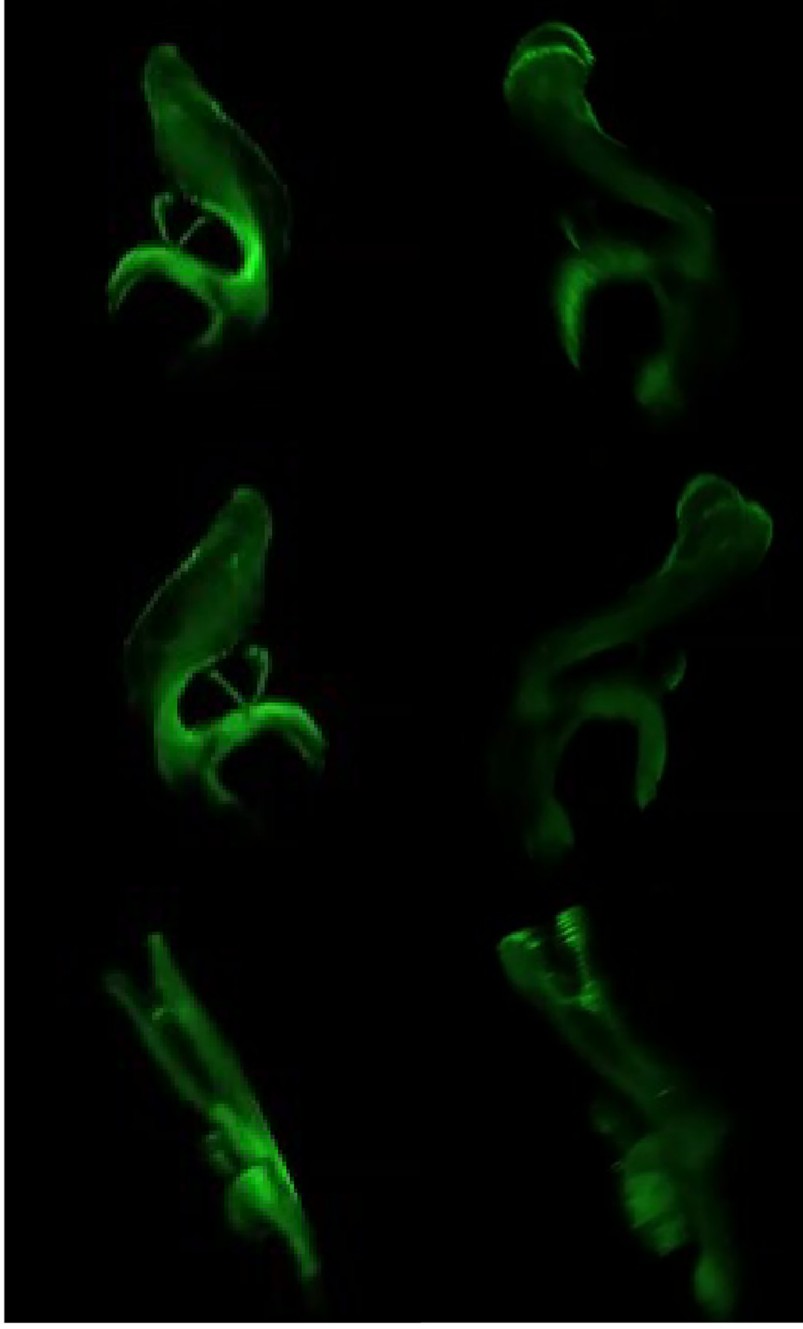

*M. projectans affinis 2*
*M. projectans* affinis 3

*M. projectans* affinis 1

**Fig 3. Images of the aedeagus structures reported for the different species of the *M. projectans* complex as visualized by confocal miscroscopy.** (A) Type I, encountered in *M. projectans* affinis 2 and *M. projectans* affinis 3; (B) Type II, detected for *M. projectans* affinis 1. From top to bottom: lateral, dorsolateral, and ventral views. Images (A) of sample 584A35 and (B) of sample 584A39 (see S7 Table for more details).

*projectans* affinis 1, with a longer aedeagus showing a rounded and wider apical region in the lateral view in comparison to the previous morphology (hereafter called type II) (Fig 3B).

## How and when did this diversification occur?

**Phylogenetic relationships and divergence dating.** The resulting phylogenetic gene trees supported the monophyly of the *M. projectans* complex (PP = 1), although the topology of the relationships within the complex varied among the genes. In fact, even both mitochondrial markers recovered different topologies, with COI presenting *M. projectans* affinis 1 and *M. projectans* affinis 3 as sister species (PP = 0.33), whereas COII clustered *M. projectans* affinis 1 with *M. projectans* affinis 2 (PP = 1). Conversely, both nuclear markers supported the clustering of *M. projectans* affinis 2 and *M. projectans* affinis 3 [PP = 0.94 (HB) and PP = 0.75 (AMD)] (Fig 4A–4D). This incongruence is reflected in the low support reported for the early offshoot of *M. projectans* affinis 3 in the species tree (PP = 0.50) (Fig 4E). According to this chronophylogenetic tree, the first divergence within the complex occurred ca. 17 Mya, and the next split occurred ca. 15 Mya.

## Patterns of distribution of genetic diversity and abiotic niches help to disentangle evolutionary history and suggest modes of speciation

**Genetic diversity, population structure, and demographic history.** In general, the analysis of haplotype diversity (Hd) or expected heterozygosity (He) for mitochondrial and nuclear markers, respectively, nucleotide diversity ($\pi$) and number of polymorphic sites (S) revealed higher values for all four genes for *M. projectans* affinis 2 and *M. projectans* affinis 3 than for *M. projectans* affinis 1 (Table 1). Neutrality tests showed some significantly negative values for COI, COII, COI+COII, and HB in all three species (Table 1).

Pairwise Fst indexes revealed significant values in the comparisons between species for all datasets, although the values recovered for AMD were lower than those reported for COI, COII, COI+COII, and HB (S9A–S9D Table). AMOVA tests executed with the same sequences showed that at least 51% of the total genetic variation was explained by differences among species (S10 Table). Conversely, the variation encountered among populations within the same species was very similar or even lower than that encountered within populations, especially for the nuclear markers (S10 Table).

The networks recovered for the three species using COI sequences generally showed starlike patterns, with central and more frequent haplotypes connected with peripheral and exclusive ones through a few mutational steps (Fig 1). However, there were some exceptions, such as haplotype 6 (Foz do Iguaçu/PR) in *M. projectans* affinis 1, haplotypes 4 (Teodoro Sampaio/SP) and 26 (Foz do Iguaçu/PR) in *M. projectans* affinis 2, and haplotypes 20 (Derrubadas/RS) and 29 (Pelotas/RS) in *M. projectans* affinis 3, which were separated by up to 7–17 mutational steps from their most similar haplotypes.

Although differentiation among species attained a minimum of 34 mutational steps for COI, intraspecific gene flow seems to have been maintained despite long distances. Thus, in *M. projectans* affinis 2, for example, haplotypes that were collected about 3,000 km apart, such as haplotypes 37 and 7, were differentiated by only two mutational steps (Fig 1C). Likewise, in the same species, haplotypes differing by as many as 10 mutational steps were found at the same point. Nonetheless, for *M. projectans* affinis 3, two haplogroups could be observed (Fig 1D): the first, with the most common haplotypes sampled in the west (H2 and H24) and radiating to the east, was almost restricted to the Atlantic Forest (only one individual found in the border between Atlantic Forest and Pampa (at Santa Maria) shared a haplotype from this haplogroup); and the other was almost restricted to the Pampa biome, with several haplotypes

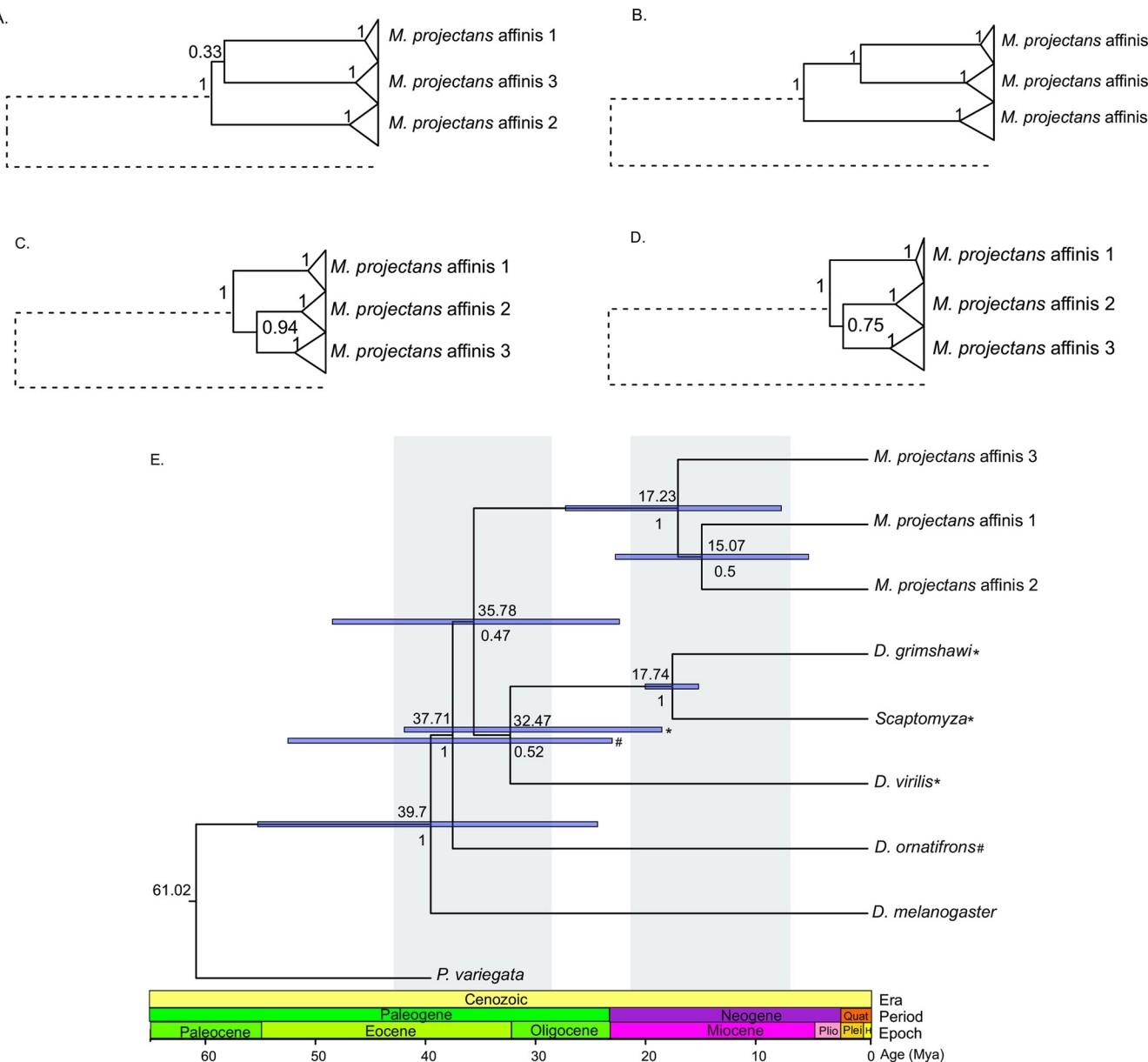

**Fig 4. Phylogenetic trees recovered by Starbeast.** (A) COI gene tree, (B) COII gene tree, (C) HB gene tree, (D) AMD gene tree, and (E) multi-locus species tree. Numbers above internal nodes in the gene trees and below the internal nodes in the species tree represent support values, as represented by the posterior probability (PP) of each clade. Numbers above internal nodes in the species tree represent mean divergence dates. Bars within internal nodes represent the confidence intervals (CI).

being sampled along the border between the Atlantic Forest and Pampa and radiating to the south. Conversely, the COI networks of *M. projectans* affinis 1 and 2 (Fig 1B and 1C) contained the most common haplotype originating from west of the Atlantic Forest, from which it radiated haplotypes to the east in *M. projectans* affinis 1 and to the east in the Atlantic Forest, to the south in the Pampa, and to the north in the Amazon in *M. projectans* affinis 2. These patterns were somewhat diluted in the COI+COII concatenated, AMD, and HB networks, due to reduced sampling, although differentiation among species remained very pronounced (S4 and S5 Figs).

According to BAPS, COI did not recover any population subdivision in any of the three species, whereas COII subdivided *M. projectans* affinis 1 and *M. projectans* affinis 3 in two subpopulations and *M. projectans* affinis 2 in three subpopulations, with two groups composed of individuals from the Atlantic Forest and another group exclusively composed of specimens from the Pampa biome (S6 Fig). Furthermore, the results provided by the Mantel Tests (S11 Table) revealed significant correlations between genetic and geographic distances only for *M. projectans* affinis 1 (rCOI = 0.770, p = 0.035) and *M. projectans* affinis 3 (rCOI = 0.398, p = 0.002; rCOII = 0.437, p = 0.026; rCOI+COII = 0.424, p = 0.029).

The results of the Extended Bayesian Skyline (EBSP) analysis agreed with the neutrality tests and the star-like patterns seen for the COI networks. In fact, EBSP suggests that *M. projectans* affinis 2 and *M. projectans* affinis 3 experienced an expansion of population size, but in different magnitudes and moments, initiated around 200 and 400 Kya, respectively (S7 Fig). According to the 95% HPD sum(indicators.alltrees) parameter, the number of effective population size changes in *M. projectans* affinis 3 and *M. projectans* affinis 2 ranged between 1 and 3 (median 1) and 1 and 3 (median 2), respectively. Nevertheless, for *M. projectans* affinis 1, as 0 was within the 95% HPD range of the sum(indicators.alltrees) parameter, EBSP did not provide significant evidence of changes in effective population size.

**Environmental distribution patterns and ecological requirements.** Currently, the three species of the *M. projectans* complex are sympatric in an area encompassing the southern region of Brazil (Fig 1, S1 Table), but some species of the complex have been collected in tropical and subtropical areas from the Neotropical Region. In the sampled areas, species of the complex were found to be associated with fungal species of *Ganoderma*, *Polyporus*, and *Trametes* (S12 Table), and at least twice, all three species were found in syntopy. In these cases, the triad was simultaneously collected using *Ganoderma* and another unidentified fungi, at Derrubadas and Foz do Iguaçu, respectively, two conserved areas of the Atlantic Forest. *Mycodrosophila projectans* affinis 2 and *M. projectans* affinis 3 also co-occurred in species of *Ganoderma* and *Polyporus* at Bossoroca and Derrubadas, respectively (S12 Table).

ENM for all three species showed AUC values higher than 0.809 (S13 Table). These models suggested the range of suitable areas differs among the three species, with present and past overlapping ranges in Southern Brazil (Fig 5). Nevertheless, suitability of the sampling area seems to have decreased progressively during the Pleistocene, reaching the lowest levels during the LGM for all three species.

Although the climatic variable that contributed most to the models in *M. projectans* affinis 2 and *M. projectans* affinis 3 was the precipitation of the driest quarter (Bio-17), for *M. projectans* affinis 1, the precipitation of the coldest quarter (Bio-19) was the most important variable (S14 Table). Measures of abiotic niche overlap based on ENM results were higher in the comparisons involving *M. projectans* affinis 1 and *M. projectans* affinis 2 (D = 0.904, I = 0.994), than when any of these were compared to *M. projectans* affinis 3 (D = 0.549, I = 0.839; D = 0.528, I = 0.858, respectively) (S15 Table). However, the null hypothesis of niche equivalency could not be rejected in any of the pairwise comparisons.

## Discussion

This study revealed an additional example of cryptic diversity for *Mycodrosophila*, which was already hypothesized as having several cryptic species [28, 89]. Although the application of DNA barcoding was fundamental to disentangling the *M. projectans* species complex, caution is necessary when interpreting results obtained solely with COI, since mitochondrial markers may suffer from the interference of numts or even hitchhiking due to maternally inherited endoparasites [31]. The influence of both factors on the proposed inclusion of three species in

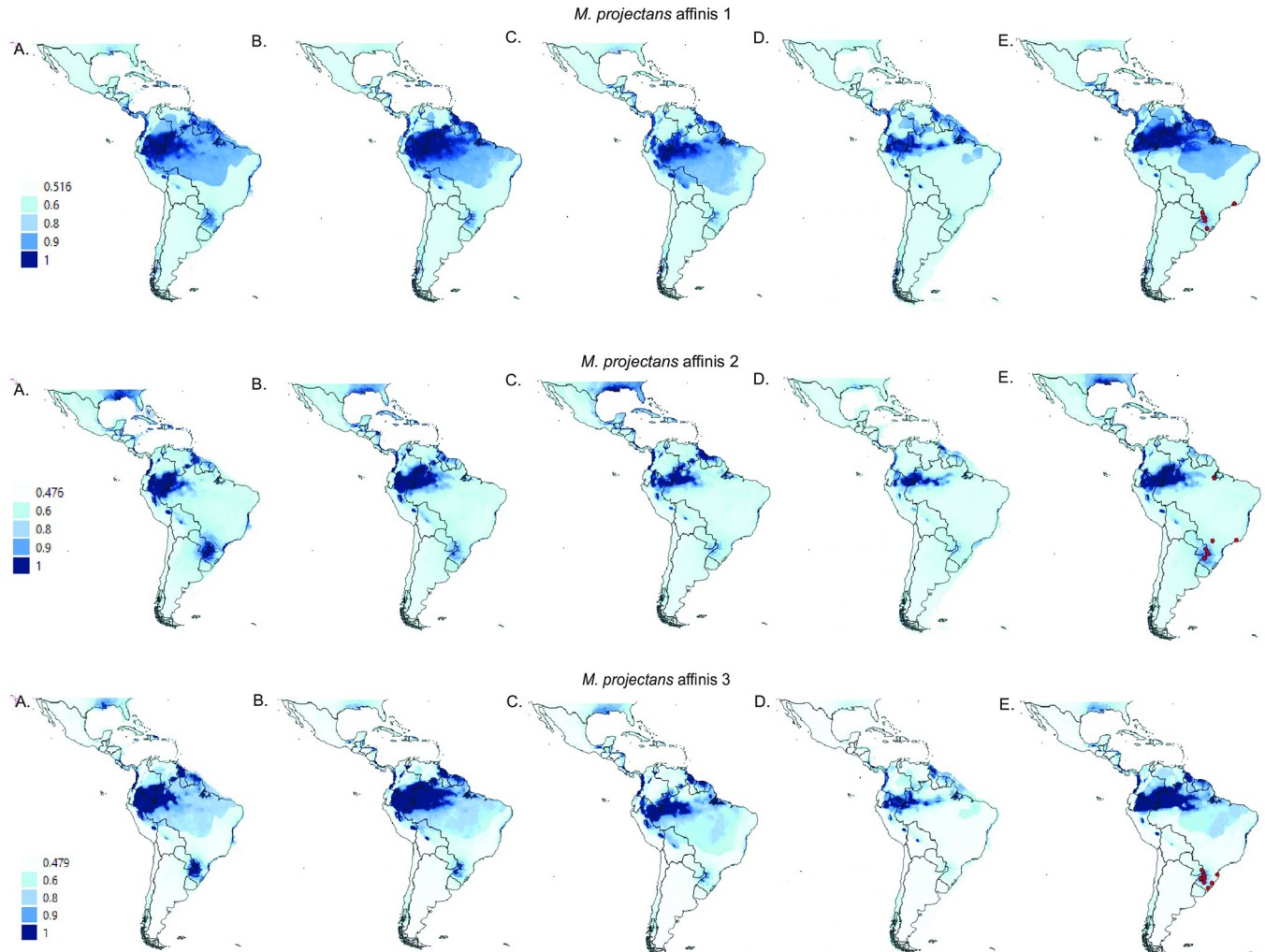

**Fig 5.** Environmental distribution models predicted by maximum entropy for each of the three species of the *M. projectans* complex for the (A) Pliocene, (B) Pleistocene (787 kya), (C) Last Interglacial, (D) Last Glacial Maximum, (E) Present. Neotropical shapefile was obtained from Löwenberg-Neto [108], as available at http://purl.org/biochartis/neo2014shp.

the *M. projectans* complex [28] can be discounted by the evidence presented in the current study.

Concerns related to the co-amplification of numts were diminished by the absence of stop codons, frameshifts, or high levels of divergence for COI sequences, patterns associated with neutral evolution that are commonly observed in paralogous copies of mitochondrial genes [41]. In addition, because numts generally cover a small portion of the mitochondrial genome [30], the congruence between the results obtained for COI and COII can be considered additional evidence against the presence of such artefacts here. In fact, for both COI and COII, we showed that reciprocal monophyly was consistent with the presence of a gap between the maximum intraspecific distances and the minimum interspecific divergences. In addition, each of the tree species presented at least 11 diagnostic characters that seemed to be fixed for each mitochondrial marker. Finally, for both markers, AMOVA indicated that more than 80% of the variation occurred among the three haplogroups, whose pairwise Fst values were close to 1.0.

In addition to the congruent results recovered by COI and COII, the nuclear markers AMD and HB also supported reciprocal monophyly of three or at least two of the lineages, respectively. The use of both the ABGD algorithm and coalescence-based Bayesian analysis, using single or multi-locus strategies, reinforced the subdivision of the *M. projectans* complex into at least three clusters, with an exact correspondence to the three lineages previously defined using DNA barcoding. These results allowed us to reject the hypothesis that the clades observed by Machado et al. [28] would be a result of selective scanning associated with three different and recent invasions of *Wolbachia* in *M. projectans* [31]. Nevertheless, it is still possible that such patterns are the result of ancient invasions independently fixed in different populations, which were first associated with cytoplasmic incompatibility and post-zygotic isolation, and later enhanced the emergence or pre-zygotic isolation through reinforcement [89–91]. In this case, *Wolbachia* could act as a pro-speciation factor, and such a longstanding infection could result in fixation also for nuclear alleles. In fact, there is evidence of *Wolbachia* infection in at least one species of the *Zygothrica* genus group, and this endosymbiont was shown to be more frequent in mycophagous than non-mycophagous Diptera [32]. Although simple diagnostic amplification could be used to confirm the presence of *Wolbachia* in our samples, shortages related to the amplification success of some of our markers showed that such estimates of infection rates would not be precise. Thus, it remains to be further evaluated whether *Wolbachia* was involved in the diversification of the *M. projectans* complex. Even so, this uncertainty does not affect the conclusion about the presence of at least three species under the general morphotype previously ascribed to *M. projectans*.

The cryptic morphology among the three species was confirmed after rigorous analysis, and only a few subtle morphological differences were found for the three species in terms of aedeagus morphology and abdominal color patterns. However, none of these allowed the three species of the complex to be unambiguously distinguished. In addition to this cryptic morphology, the *M. projectans* complex also detaches for recurrent sympatry and syntopy among its species, as observed here in three samples from the Atlantic Forest and the Pampa biome. The coexistence of cryptic species seems to be a common pattern in *Mycodrosophila* since Lacy [92] also reported syntopy for two cryptic species in the Nearctic (*M. claytonae* A and B). Within the *Zygothrica* genus group, the co-occurrence of cryptic species has also been reported for Neotropical species of the genera *Zygothrica* and *Hirtodrosophila* [28, 93]. Nevertheless, for the *M. projectans* complex, beyond co-occurrence, patterns of biotic and abiotic niches were also quite similar, and species used the same genera of fungi as breeding sites. Despite differences among species concerning the amplitude of potential distribution, neither pairwise comparison between abiotic niches recovered through ENM strategies rejected the null hypothesis of niche equivalency. These results support the hypothesis of niche conservatism for the *M. projectans* complex [29], suggesting that it may be a common pattern in the *Zygothrica* genus group. Nevertheless, further ecological studies are necessary to clarify how these species are using their resources and how competition affects their co-occurrence.

Patterns of morphological similarity and niche equivalency seem to have been maintained despite ancient divergence. In this sense, although there were incongruences among gene trees regarding the topology of the relationships within the *M. projectans* complex, the multi-locus strategy used to reconstruct the species tree under a relaxed molecular clock strategy allowed us to date the two cladogenesis events to approximately 17–15 Mya ago. These estimates indicated that the diversification of the *M. projectans* complex occurred in the Neogene period during the Miocene [94]. This contrasts with the pattern commonly found for Drosophilidae, in which cryptic species encompass recently diverged lineages [93, 95, 96] that did not have enough time to accumulate diagnosable features. Thus, this study provides an interesting example of morphological conservation, despite ancient divergence. Furthermore, this ancient

divergence, in view of the similarity of niches evidenced here for the three lineages, allows us to hypothesize that divergence of the species occurred in allopatry or parapatry, with current records of sympatry and syntopy reflecting secondary contact. This hypothesis of allo- or parapatric distribution in more ancient times was also supported by EBSP analyses, which showed contrasting patterns of demographic fluctuations for all species. In fact, whereas *M. projectans* affinis 1 did not present significant signals of population expansion, such signals were supported for *M. projectans* affinis 2 and *M. projectans* affinis 3 as occurring at different moments of the Middle Pleistocene [94], ca. 200 and 400 Kya, respectively. Such differences would certainly not be expected if species were sympatric, especially in the face of straightforward evidence of niche equivalence.

Furthermore, these three species seem to have expanded to Southern Brazil quite recently, as shown by the ENMs projected for different moments of the past. The influence of precipitation on the modelling strategies further supported the interference of the climatic oscillations of the Quaternary in the population dynamics of these species, since this period is characterized by alternating periods of dry and wet conditions in the Southern Hemisphere [97, 98]. These results are also concordant with phylogeographic studies performed with other Brazilian drosophilids, which also showed signals of population expansion dated to the Pleistocene [99–101]. This general pattern suggests that several drosophilid species have been subject to demographic oscillations during the glacial–interglacial cycles of this Epoch [99, 102]. However, although for cactophilic species, expansions are frequently dated to glacial periods [99], the effect of glaciations was the opposite for other species, as recently supported for *D. maculifrons* and *D. ornatifrons* [100, 101]. The pattern found here for the *M. projectans* complex adds three more species to the list of Drosophilidae species suggesting recent population expansion in southern Brazil. Although further sampling is necessary to clarify this scenario, especially in other distribution areas, such results certainly help to disentangle some complexities related to the evolution of the *M. projectans* complex.

Even with this recent diversification, some species of the *M. projectans* complex showed significant levels of population structure between the two southernmost Brazilian biomes: the Pampa and Atlantic Forest. The levels of nucleotide and haplotype or allele diversity seen here for the *M. projectans* complex were also considerably higher than those previously found for several other Drosophilidae species [99, 100, 102–106]. In fact, nucleotide diversity for *M. projectans* affinis 3 reached more than 10,000 times the value found for *D. maculifrons*, a species that also showed signals of population expansion dated to the LGM [100]. This high diversity could be an intrinsic characteristic of the *M. projectans* complex and possibly of other *Mycodrosophila* species, suggesting the occurrence of high levels of gene flow in response to the restrictions imposed by the ephemerality of resources [24, 92]. In this sense, several cases in which haplotypes from different locations were much more similar in the number of mutational steps than some haplotypes found in the same locality were observed in the haplotype networks of *M. projectans* affinis 2. This suggests metapopulation dynamics, as proposed for other Neotropical species of *Drosophila* in response to cyclic fluctuations in the availability of resources [107]. Even so, for *M. projectans* affinis 1 and *M. projectans* affinis 3, the Mantel tests recovered a significant correlation between geographical distances and genetic divergences, suggesting the application of an isolation-by-distance pattern of population differentiation.

## Conclusion

In conclusion, this study supports the existence of at least three cryptic species within the *M. projectans* complex, as initially proposed by Machado et al. [28]. As congruent results were obtained for different markers and techniques, it was possible to exclude numts and recent

hitchhiking with endosymbiont parasites. Despite diversification of these lineages dated to around 17–15 Mya (Miocene), only subtle morphological differences were encountered among species, and abiotic niches were equivalent among them. Although the three species seem to have expanded to the sampled region after the LGM, they showed high levels of genetic diversity, and in some cases, some level of population structure between Biomes. Despite inherent difficulties in sampling these species, this study represents a step forward in the understanding of potential spatiotemporal and ecological patterns related to their diversification.

## Supporting information

**S1 Fig. Illustration of the right wing of a specimen of the *M. projectans* complex showing the landmarks chosen for morphometry analysis.**
(PDF)

**S2 Fig.** Majority rule consensus tree recovered through Bayesian analysis based on (A) COI, (B) COII, (C) HB, and (D) AMD datasets. Branch lengths are proportional to the scale, given in substitutions per site. Values near the internal branches represent the posterior probability (PP) of each clade.
(PDF)

**S3 Fig. Cartesian plot from wings relative warps reported for each of the three species of the *M. projectans* complex.**
(PDF)

**S4 Fig. Median-joining network reconstructed with COI+COII haplotypes characterized for each of the three species of the *M. projectans* complex.** (A) *M. projectans* affinis 1, (B) *M. projectans* affinis 2, and (C) *M. projectans* affinis 3. Each circle represents a different haplotype, whose size is proportional to frequency. Each color represents different sampling points, in accordance with the legend presented on Fig 1. Black small circles represent median vectors. Dashes in the lines connecting different haplotypes represent the number of mutations between them.
(PDF)

**S5 Fig.** Median-joining network reconstructed with (A) HB, and (B) AMD phased alleles characterized for each of the three species of the *M. projectans* complex. Each circle represents a different haplotype, whose size is proportional to frequency. Each color represents different sampling points, in accordance with the legend presented on Fig 1. Black small circles represent median vectors. Dashes in the lines connecting different haplotypes represent the number of mutations between them.
(PDF)

**S6 Fig.** Results from the spatial clustering analysis performed by BAPS for each of the three species of the *M. projectans* complex based on (A) COI, and (B) COII datasets. Axis x and y reflect the longitude and latitude of each population, whereas each color represents a different cluster of populations.
(PDF)

**S7 Fig.** Extended Bayesian Skyline Plots depicting oscillations of population sizes faced by *M. projectans* affinis 2 (A) and *M. projectans* affinis 3 (B) during the last 1 Mya.
(PDF)

**S1 Table. List of sampling sites of the *M. projectans* complex, with their respective geographic coordinates.**
(XLSX)

**S2 Table. List of specimens used in this study, with their respective origin and GenBank accession numbers.**
(XLSX)

**S3 Table. Description of wing landmarks used in morphometric analysis.**
(XLSX)

**S4 Table. Results of pairwise comparisons involving Dn and Ds, and the codon-based Z test of purifying selection performed to assist in the detection of numts for the two mitochondrial datasets.**
(XLSX)

**S5 Table. Number of clusters recovered for the *M. projectans* complex by ABGD, GMYC, and Stacey, under different datasets.**
(XLSX)

**S6 Table. List of diagnostic characters recovered for each of the three species of the *M. projectans* complex under the COI, and COII datasets.** Sites that did not reveal diagnostic nucleotides were omitted.
(XLSX)

**S7 Table. Patterns of morphology found in different specimens of the *M. projectans* complex based on the evaluation of abdominal spots and aedeagus type.**
(XLSX)

**S8 Table. Results of ANOVA and CVA of wing variation among the three species of the *M. projectans* complex.**
(XLSX)

**S9 Table. Pairwise values of Fst among the three species of the *M. projectans* complex, as calculated for different datasets.**
(XLSX)

**S10 Table. Results of the Analyses of Molecular Variance (AMOVA) considering the subdivision of the *M. projectans* complex in three species, as recovered with the use of different datasets.**
(XLSX)

**S11 Table. Results of the Mantel tests performed for each of the three species of the *M. projectans* complex with the use of different datasets.**
(XLSX)

**S12 Table. List of resources in which each of the three species of the *M. projectans* complex was collected.** Sampling point number refers to codes presented on S1 Table.
(XLSX)

**S13 Table. Number of points and mean AUC values presented by the ENM reconstructed for each of the three species of the *M. projectans* complex.**
(XLSX)

**S14 Table. Estimates of relative contributions of each of the employed environmental variables to the ENMs reconstructed for each of the three species of the *M. projectans* complex.**
(XLSX)

**S15 Table.** Pairwise values of abiotic niche overlap estimated among the three species of the *M. projectans* complex according to Schoener's D (D) (above diagonal) and Hellinger's (I) (below diagonal) indexes.
(XLSX)

## Acknowledgments

We are grateful to the authorities responsible for Brazilian Ecological Stations, Reserves or Parks and to the private properties owners that allowed us to perform Drosophilidae searches in their area. All these collections were authorized by the Brazilian Ministério do Meio Ambiente (MMA), in the form of the Sistema de Autorização e Informação em Biodiversidade (SISBIO), and we thank them for their commitment to ethical guidelines and conservation efforts concerning Brazilian scientific activities. We would also like to thank to Dr. Eduardo Bernardi for the fungi identification and to M.Sc. João Pedro Junges dos Santos for his help in drosophilids identification.

## Author Contributions

**Conceptualization:** Stela Machado, Maiara Hartwig Bessa, Lizandra Jaqueline Robe.

**Investigation:** Stela Machado, Maiara Hartwig Bessa, Marco Silva Gottschalk, Lizandra Jaqueline Robe.

**Methodology:** Stela Machado, Maiara Hartwig Bessa, Bruna Nornberg, Marco Silva Gottschalk, Lizandra Jaqueline Robe.

**Project administration:** Lizandra Jaqueline Robe.

**Supervision:** Lizandra Jaqueline Robe.

**Writing – original draft:** Stela Machado, Maiara Hartwig Bessa, Lizandra Jaqueline Robe.

**Writing – review & editing:** Maiara Hartwig Bessa, Marco Silva Gottschalk, Lizandra Jaqueline Robe.

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
