## [Decision Letter · Decision Letter 0]

25 Oct 2021

PONE-D-21-28403Unveiling the Mycodrosophila projectans (Diptera, Drosophilidae) species complex: insights into the evolution of three Neotropical cryptic and syntopic speciesPLOS ONE

Dear Dr. Robe,

Thank you for submitting your manuscript to PLOS ONE. After careful consideration, we feel that it has merit but does not fully meet PLOS ONE’s publication criteria as it currently stands. Therefore, we invite you to submit a revised version of the manuscript that addresses the points raised during the review process.

Four external referees and me have now reviewed your manuscript and all have pointed out a number of specific issues that need addressing in a revised version of this manuscript. I am sorry this review process has been longer than anticipated, but some reviewers needed extra time to handle this assignment. I agree that the problems pointed out by the reviewers need to be specifically addressed, and I also noticed a number of grammar and English language problems throughout the MS. I strongly urge the authors to have an English speaker edit this MS before resubmission.  Presentation in correct English is a requirement of this journal. Also please be sure to always number the pages of submitted manuscripts and include line numbering for the reviewers.

We look forward to receiving your revised manuscript.

Kind regards,

William J. Etges

Academic Editor

PLOS ONE

Journal Requirements:

2. We note that you are reporting an analysis of a microarray, next-generation sequencing, or deep sequencing data set. PLOS requires that authors comply with field-specific standards for preparation, recording, and deposition of data in repositories appropriate to their field. Please upload these data to a stable, public repository (such as ArrayExpress, Gene Expression Omnibus (GEO), DNA Data Bank of Japan (DDBJ), NCBI GenBank, NCBI Sequence Read Archive, or EMBL Nucleotide Sequence Database (ENA)). In your revised cover letter, please provide the relevant accession numbers that may be used to access these data. For a full list of recommended repositories, see http://journals.plos.org/plosone/s/data-availability#loc-omics or http://journals.plos.org/plosone/s/data-availability#loc-sequencing.

“YES - This study was supported by Universal-CNPq 14/2013, process numbers 471174/2013-0 and 472973/2013-4.”

4. Thank you for stating the following in the Funding Section of your manuscript:

“This study was supported by Universal-CNPq 14/2013, process numbers 471174/2013-0 and 472973/2013-4.”

We note that you have provided additional information within the Funding Section. Please note that funding information should not appear in other areas of your manuscript. We will only publish funding information present in the Funding Statement section of the online submission form.

““YES - This study was supported by Universal-CNPq 14/2013, process numbers 471174/2013-0 and 472973/2013-4.””

5. We note that Figures 1A and 5 in your submission contain map images which may be copyrighted. All PLOS content is published under the Creative Commons Attribution License (CC BY 4.0), which means that the manuscript, images, and Supporting Information files will be freely available online, and any third party is permitted to access, download, copy, distribute, and use these materials in any way, even commercially, with proper attribution. For these reasons, we cannot publish previously copyrighted maps or satellite images created using proprietary data, such as Google software (Google Maps, Street View, and Earth). For more information, see our copyright guidelines: http://journals.plos.org/plosone/s/licenses-and-copyright.

    1. You may seek permission from the original copyright holder of Figure(s) [#] to publish the content specifically under the CC BY 4.0 license. 

Maps at the CIA (public domain): https://www.cia.gov/library/publications/the-world-factbook/index.html and https://www.cia.gov/library/publications/cia-maps- publications/index.html

Reviewers' comments:

Reviewer's Responses to Questions

**Comments to the Author**

1. Is the manuscript technically sound, and do the data support the conclusions?

Reviewer #1: Yes

Reviewer #2: Yes

Reviewer #3: Partly

Reviewer #4: Yes

2. Has the statistical analysis been performed appropriately and rigorously? 

Reviewer #1: Yes

Reviewer #2: Yes

Reviewer #3: I Don't Know

Reviewer #4: Yes

3. Have the authors made all data underlying the findings in their manuscript fully available?

Reviewer #1: Yes

Reviewer #2: Yes

Reviewer #3: Yes

Reviewer #4: Yes

4. Is the manuscript presented in an intelligible fashion and written in standard English?

Reviewer #1: Yes

Reviewer #2: Yes

Reviewer #3: Yes

Reviewer #4: Yes

5. Review Comments to the Author

Reviewer #1: Unveiling the Mycodrosophila projectans (Diptera, Drosophilidae) species complex: insights into the evolution of three Neotropical cryptic and syntopic species

Stela Machado, Maiara Hartwig Bessa, Bruna Nornberg, Marco Silva Gottschalk, Lizandra Jaqueline Robe

Summary:

The authors show that the species Mycodrosophila projectans consists of (at least) three cryptic species. They do so by using DNA barcode sequencing and excluding interferences from numts and hitchhiking mitochondrial DNA in endosymbiotic parasites. The authors found a surprizing result that, despite the three species being so similar and overlapping in their niches, they have diverged in two events, dating back 15 and 17 MYA. The authors offer an explanation that the divergence was soon followed by geographic isolation, and that these species later came back in contact through migration events, possibly driven by the scarcity of their mushroom hosts. The authors also found subtle morphological characters that can be used to distinguish the three species from each other, which can be useful for future field research with these species.

Critique:

1) This is a thorough study, using many analyses and techniques to understand that there are three species present, how they can be distinguished on the molecular levels and identifying morphological markers, leading to the identification of key features on the abdominal color pattern and the male genitalia. Although I have not performed these kinds of analyses myself, the results are, at least to me, convincingly demonstrated and explained. I like how the authors have tried to figure out the timing of divergence and the discrepancy between the species being comparatively old but still cryptic. It makes a nice story.

2) When I read through the paper, I was a bit lost until I got to the discussion, where finally everything was nicely explained. I think that the paper could gain some clarity in the results section if the results were not just written out, but if each paragraph could have the following logic, beginning with stating the question, then stating what you did, then what happened (this is what the Results section currently contains), followed by a brief statement what the results mean and stating what the next question therefore is. This pattern of writing gives a reader not so familiar with the methodology more clarity, especially important for the broad audience of PLoS ONE.

3) There were some minor punctuation, phrasing and other language errors, especially in the first half of the paper, which the editors can correct later in the process. I just want to point out one: cactophilic is spelled with an i after the letter h.

4) Otherwise well done!

Reviewer #2: I think your manuscript convincingly shows your samples of M. projectans consist of three reciprocally monophyletic species. Below are my comments on the manuscript. In general, there three issues that need to be addressed. First, the grammatical errors should be corrected. I think I have found most of them (see below) but the manuscript should be re-checked. Second, the discrepancy in sequencing samples across species (and loci) should be explained in the text. The discrepancy would likely not affect conclusions, but the method of choosing samples should be described. Were samples for sequencing randomly chosen from each geographic location? Was it based on tentative morphological species delimitation? Was it simply all males collected? Why the discrepancy between mtDNA and nucDNA sample sizes? Was there failure of amplification for some specimens? And three, Wolbachia is importantly discussed as a factor that could disrupt reciprocal monophyly of species, and I think the evidence presented convincingly shows this is not the case for these species. However, I think a simple PCR diagnostic for Wolbachia infection is warranted here to support this conclusion. Is there a known Wolbachia infection in these species? At the very least, there should be some discussion as to what is known about Wolbachia incidence and/or prevalence in these or closely related species. There are several additional minor comments included among the comments below.

Apologies, but the copy I was provided with did not have page or line numbers. P# refers to the paragraph number for the given section.

Intro:

P1: “ mechanisms responsible for these patterns . . .” -> “mechanism responsible for this pattern” (i.e. cryptic variation)

P2: “using them as resource . . .”. -> “using them as a resource”

P4: Machado et. al is said to describe the same external morphotype for the projectans complex, then the next sentence describes the limitation of mtDNA markers. It should be explicitly stated that Machado et. al used a mtDNA marker to delimit species.

P5: “These strategies allowed confirmation the existence . . .” -> “confirmation of the existence . . .”

Methods:

Table 1: Was the reduced sampling of nuclear loci (and disparity between species) by choice or failure of amplification? A quick glance at table S2 makes it look like sequencing was guided by geographic location. This would likely not affect overall conclusions of the paper, but it should be addressed in the text.

Morphological and molecular approaches for species delimitation:

P3: How was convergence of the MCMC used with the Stacey analysis assessed?

Results:

Species delimitation by molecular approaches:

P1: “None coding sequences” -> “No coding sequences”

P2: “revealed reciprocally monophyletic.” -> “revealed to be reciprocally monophyletic”

Genetic Diversity, structure and demographic history:

P4: “Likewise, at the same . . . found in the same point” -> “Likewise, in the same . . . found at the same point.”

“on west . . . radiating to east.” -> “in the west . . . radiating to the east.”

“originating from west of the Atlantic forest . . . northwest to Pampa”. This is not clear. The only Pampa locality is south of all Atlantic forest collection sites.

P5: “. . . did not recover none population . . .” -> “. . . did not recover any population . . .”

P6: “In fact, EBPS . . .” -> “In fact, EBSP . . .”

Environmental distribution patterns and ecological requirements

P2: “. . . overlaps at the Southern Brazil . . .” -> “. . . overlaps in Southern Brazil . . .”

P3: “ . . . in any of these cases the null hypothesis of niche equivalence could be rejected”. This is not what is stated in the Discussion, or what Table S15 shows. It should state that the null hypothesis of niche equivalency could not be rejected.

Discussion:

P3 (and general comment about Wolbachia throughout the manuscript): The first sentence here (“Additionally, the congruence . . .”) needs more explanation. It is not clear what is meant by “selective scanning assoc. with three Wolbachia invasions”. If three different Wolbachia invasions fixed in the sufficiently distant past (your analyses do suggest relatively ancient lineage divergence), I’m not sure how your results would look any different. I guess I’m just not clear on what is being described here.

Only VERY recent Wolbachia invasions (i.e. not yet fixed in the invaded population) or maintenance of intermediate Wolbachia frequencies would remove a signal of reciprocal monophyly in the mtDNA. However, fixed Wolbachia invasions (anytime after lineage divergence) could produce discordant mtDNA vs. nucDNA topologies (in addition to significant reductions in tajima’s D in mtDNA loci, depending on timing), which your analyses show. A more general comment: Do these species have Wolbachia? I like that the Wolbachia is discussed as a potential factor that could disrupt species delimitations, but there is nothing presented as to whether Wolbachia even exists (or existed) in this clade. A simple PCR diagnostic on some of your samples is certainly warranted here. Or at the very least a few sentences describing what is known about the presence of Wolbachia (and their associated phenotypes, if known) in these or closely related species. All of this is to say that your primary conclusion is not affected (you still convincingly have three reciprocally monophyletic species) but I think there is certainly a possibility that Wolbachia (or other endosymbionts) could have influenced some of the phylogenetic and population genetic patterns observed.

P5: “ . . . M. projectans affinis 2 presents higher suitability for this region [Dominican Republic].” This is very difficult to see in the figure. Is there a way to easily quantify this that shows higher suitability relative to the other species?

P8: “. . . seems to have been hold despite . . .” -> “ . . . seems to have been maintained despite . . .”

” . . . allows us hypothesizing . . .” -> “ . . . allows us to hypothesize . . .”

Reviewer #3: I think the biological problem is interesting, but the sample size to answer the biological question is not significant. The sampled area is restricted to interior south Atlantic Forest and Pampa and COI mt sequences are predominant. I think the authors have data to elaborate hypotheses about diversification patterns for the group but not to discuss process.

Reviewer #4: Review of Machado et al., Unveiling the Mycodrosophila projectans (Diptera, Drosophilidae) species complex: insights into the evolution of three Neotropical cryptic and syntopic species

Summary

This paper examines the phylogeography of the Mycodrosophila projectans species complex, a mycophagous lineage in the Zygothrica Genus Group. The authors have thoroughly analyzed a combination of mitochondrial and nuclear gene data to discover three cryptic species in this complex. Overally, I found this a very easy paper to read and review. The authors did a wonderful job explaining all of their analyses. I only have a couple minor comments about the dating analyses.

• How did you select the outgroup taxa for this study? This isn’t really justified in the paper and the species selected seem like a random mix of taxa across the phylogeny, with a slight focus on the Hawaiian Drosophila-Scaptomyza lineage. Was this done to facilitate dating analyses? The Tamura and Survov dates are quite different from one another. Perhaps this might warrant some discussion? Obbard et al. (2012) do discuss a wide range of dates and it might be helpful to refer to this paper in your discussion.

6. PLOS authors have the option to publish the peer review history of their article (what does this mean?). If published, this will include your full peer review and any attached files.

Reviewer #1: No

Reviewer #2: No

Reviewer #3: No

Reviewer #4: **Yes: **Patrick M O'Grady

---

## [Author Response · Author response to Decision Letter 0]

5 Jan 2022

Dear Dr. William J. Etges,

Please find attached the MS entitled “Unveiling the Mycodrosophila projectans (Diptera, Drosophilidae) species complex: insights into the evolution of three Neotropical cryptic and syntopic species” which contains a revised version of the previous submission to PLOS ONE (PONE-D-21-28403). We thank you for the opportunity as well as for the constructive criticism from reviewers, which allowed several improvements in our study.

We followed carefully the recommendations and provide below a point-by-point explanation (in bold) on how we proceeded about each specific comment of the reviewers (in italics). As recommended, we also provide a revised manuscript and an additional manuscript file where all the changes made are highlighted.

Concerning deposition of data in repositories, all new sequences were submitted to GenBank, and accession numbers are provided in Table S2. These sequences will be directly released once they appear on a published document, according to NCBI standards.

The Neotropical shapefile employed on Figures 1 and 5 was created by Löwenberg-Neto (2014), which is accordingly cited on the figure’s legends. According to Löwenberg-Neto (2014), this “shapefile is freely available and may be downloaded at http://purl.org/biochartis/neo2014shp”. The Brazilian Biomes shapefiles employed on Figure 1 are the result of a cooperation signed between the Instituto Brasileiro de Geografia e Estatística (IBGE) and the Ministry of the Environment, two Brazilian government entities. It is also freely available and may be downloaded at https://www.ibge.gov.br/geociencias/cartas-e-mapas/informacoes-ambientais/15842-biomas.html?=&t=downloads. 

We are confident the current version is much improved and hope it reaches the level of quality expected by PLOS ONE.

Sincerely,

L. J. Robe, on behalf of all authors

Reviewer 1:

Reviewers Comments to the Author

The authors show that the species Mycodrosophila projectans consists of (at least) three cryptic species. They do so by using DNA barcode sequencing and excluding interferences from numts and hitchhiking mitochondrial DNA in endosymbiotic parasites. The authors found a surprizing result that, despite the three species being so similar and overlapping in their niches, they have diverged in two events, dating back 15 and 17 MYA. The authors offer an explanation that the divergence was soon followed by geographic isolation, and that these species later came back in contact through migration events, possibly driven by the scarcity of their mushroom hosts. The authors also found subtle morphological characters that can be used to distinguish the three species from each other, which can be useful for future field research with these species.

Critique:

1) This is a thorough study, using many analyses and techniques to understand that there are three species present, how they can be distinguished on the molecular levels and identifying morphological markers, leading to the identification of key features on the abdominal color pattern and the male genitalia. Although I have not performed these kinds of analyses myself, the results are, at least to me, convincingly demonstrated and explained. I like how the authors have tried to figure out the timing of divergence and the discrepancy between the species being comparatively old but still cryptic. It makes a nice story.

2) When I read through the paper, I was a bit lost until I got to the discussion, where finally everything was nicely explained. I think that the paper could gain some clarity in the results section if the results were not just written out, but if each paragraph could have the following logic, beginning with stating the question, then stating what you did, then what happened (this is what the Results section currently contains), followed by a brief statement what the results mean and stating what the next question therefore is. This pattern of writing gives a reader not so familiar with the methodology more clarity, especially important for the broad audience of PLoS ONE.

3) There were some minor punctuation, phrasing and other language errors, especially in the first half of the paper, which the editors can correct later in the process. I just want to point out one: cactophilic is spelled with an i after the letter h.

4) Otherwise well done!

We thank the reviewer for the motivating comments as well as for the suggestions to improve our manuscript. We have stated the initial question involving each of the different approaches on the Results section. We also corrected some grammar errors and misspellings that passed on the first version of the manuscript. We apologize for them.

Reviewer 2:

Reviewers Comments to the Author

I think your manuscript convincingly shows your samples of M. projectans consist of three reciprocally monophyletic species. Below are my comments on the manuscript. In general, there three issues that need to be addressed. 

 → First, the grammatical errors should be corrected. I think I have found most of them (see below) but the manuscript should be re-checked. 

We appreciate your suggestions to improve our manuscript. The grammatical errors were checked and corrected. Thank you for your detailed review on this aspect.

→ Second, the discrepancy in sequencing samples across species (and loci) should be explained in the text. The discrepancy would likely not affect conclusions, but the method of choosing samples should be described. Were samples for sequencing randomly chosen from each geographic location? Was it based on tentative morphological species delimitation? Was it simply all males collected? Why the discrepancy between mtDNA and nucDNA sample sizes? Was there failure of amplification for some specimens? 

These discrepancies are now addressed on the manuscript, where we added the following sentences:

1) Lines 136-138: “Just males were employed in further analyses because morphological differences in Drosophilidae are frequently described or even found only through inspection of the internal genital structures of males.”

2) Lines 327-328: “Sampling disparity occurred due to frequent failures of amplification, that commonly led to the premature termination of some DNA samples.” 

→ And three, Wolbachia is importantly discussed as a factor that could disrupt reciprocal monophyly of species, and I think the evidence presented convincingly shows this is not the case for these species. However, I think a simple PCR diagnostic for Wolbachia infection is warranted here to support this conclusion. Is there a known Wolbachia infection in these species? At the very least, there should be some discussion as to what is known about Wolbachia incidence and/or prevalence in these or closely related species. There are several additional minor comments included among the comments below.

Further information about Wolbachia and further interpretation of the results were added to the text:

1) In the Introduction, lines 103-106: “Furthermore, as the endoparasite Wolbachia has already been reported in some groups of mycophagous drosophilids, including one species of Hirtodrosophila, and as several species of insects can occur in a single fungus [29], mushrooms can be considered a hotspot for horizontal transfer of this endoparasite.”

2) In the Discussion, lines 537-554: “These results allowed rejecting the hypothesis that the clades observed by Machado et al. [25] would be a result of selective scanning associated with three different and recent invasions of Wolbachia in M. projectans [28]. Nevertheless, it is still possible that such patterns are resultant of ancient invasions independently fixed in different populations, that were first associated with cytoplasmic incompatibility, and post-zygotic isolation, and later enhanced the emergence or pre-zygotic isolation through reinforcement [91-93]. In this case, Wolbachia could act as a pro-speciation factor, and such a longstanding infection could result in fixation also for nuclear alleles. In fact, there is evidence for Wolbachia infection in at least one species of the Zygothrica genus group, and this endosymbiont was shown to be more frequent in mycophagous than non-mycophagous Diptera [29]. Although at a first sight simple diagnostic amplification could be used to confirm the presence of Wolbachia in our samples, shortages related to the amplification success presented by some of the markers employed here would not allow a confident description of prevalence patterns with such a methodology. Thus, it remains to be further evaluated if Wolbachia could or not be involved in the speciation of the M. projectans complex. Even so, this uncertainty does not affect the conclusion about the presence of at least three species under the general morphotype previously ascribed to M. projectans.”

P# refers to the paragraph number for the given section.

Intro:

P1: “ mechanisms responsible for these patterns ...” -> “mechanism responsible for this pattern” (i.e. cryptic variation) 

Corrected.

P2: “using them as resource ...” -> “using them as a resource”

Corrected.

P4: Machado et. al is said to describe the same external morphotype for the projectans complex, then the next sentence describes the limitation of mtDNA markers. It should be explicitly stated that Machado et. al used a mtDNA marker to delimit species.

The sentence was changed to “However, when some of these Neotropical populations were evaluated through DNA Barcoding approaches using only nucleotide sequences of the mitochondrial cytochrome oxidase c subunit I (COI) gene, at least three different lineages were uncovered [25]: M. projectans, M. projectans affinis 1 and M. projectans affinis 2.” (lines 92-96).

P5: “These strategies allowed confirmation the existence ...” -> “confirmation of the existence ...”

This sentence was corrected to: “These strategies allowed us to confirm the existence”.

Methods:

Table 1: Was the reduced sampling of nuclear loci (and disparity between species) by choice or failure of amplification? A quick glance at table S2 makes it look like sequencing was guided by geographic location. This would likely not affect overall conclusions of the paper, but it should be addressed in the text.

The two questions are now addressed on the text:

1) The reduced sampling of nuclear markers is mostly related to amplification failures (see lines 327-328): “Sampling disparity occurred due to frequent failures of amplification, that commonly led to the premature termination of some DNA samples.”

2) Sequencing was not guided by geographic location, but by sampling success. Furthermore, only males were employed in morphological and molecular analyses, which further reduced sampling size (see lines 136-138): 

Morphological and molecular approaches for species delimitation:

P3: How was convergence of the MCMC used with the Stacey analysis assessed?

This information was added on lines 214-215: “Convergence was evaluated in Tracer 1.60 [65], checking likelihood and posterior probability traces and confirming that all ESS values are above 200.”

Results:

Species delimitation by molecular approaches:

P1: “None coding sequences” -> “No coding sequences” 

Corrected for “No sequence” since all sequences are coding.

P2: “revealed reciprocally monophyletic.” -> “revealed to be reciprocally monophyletic”

Corrected.

Genetic Diversity, structure and demographic history:

P4: “Likewise, at the same ...found in the same point” -> “Likewise, in the same ...found at the same point.”

Corrected.

“on west ...radiating to east.” -> “in the west ...radiating to the east.”

Corrected.

“originating from west of the Atlantic forest ...northwest to Pampa”. This is not clear. 

The only Pampa locality is south of all Atlantic forest collection sites.

The sentence was corrected to: “originating from the west of the Atlantic Forest… to the south in the Pampa”. Thank you for noting this mistake.

P5: “...did not recover none population ...” -> “...did not recover any population …”

Corrected.

P6: “In fact, EBPS ...” -> “In fact, EBSP...”

Corrected.

Environmental distribution patterns and ecological requirements

P2: “...overlaps at the Southern Brazil ...” -> “...overlaps in Southern Brazil ...”

Corrected.

P3: “... in any of these cases the null hypothesis of niche equivalence could be rejected”. This is not what is stated in the Discussion, or what Table S15 shows. It should state that the null hypothesis of niche equivalency could not be rejected.

Corrected to: “the null hypothesis of niche equivalency could not be rejected in any pairwise comparison”.

Discussion:

P3 (and general comment about Wolbachia throughout the manuscript): The first sentence here (“Additionally, the congruence...”) needs more explanation. It is not clear what is meant by “selective scanning assoc. with three Wolbachia invasions”. If three different Wolbachia invasions fixed in the sufficiently distant past (your analyses do suggest relatively ancient lineage divergence), I’m not sure how your results would look any different. I guess I’m just not clear on what is being described here.

Only VERY recent Wolbachia invasions (i.e. not yet fixed in the invaded population) or maintenance of intermediate Wolbachia frequencies would remove a signal of reciprocal monophyly in the mtDNA. However, fixed Wolbachia invasions (anytime after lineage divergence) could produce discordant mtDNA vs. nucDNA topologies (in addition to significant reductions in tajima’s D in mtDNA loci, depending on timing), which your analyses show. A more general comment: Do these species have Wolbachia? I like that the Wolbachia is discussed as a potential factor that could disrupt species delimitations, but there is nothing presented as to whether Wolbachia even exists (or existed) in this clade. A simple PCR diagnostic on some of your samples is certainly warranted here. Or at the very least a few sentences describing what is known about the presence of Wolbachia (and their associated phenotypes, if known) in these or closely related species. All of this is to say that your primary conclusion is not affected (you still convincingly have three reciprocally monophyletic species) but I think there is certainly a possibility that Wolbachia (or other endosymbionts) could have influenced some of the phylogenetic and population genetic patterns observed.

This reasoning is now considered in P3 (lines 532-554): “In this study, besides the congruent results recovered by both mitochondrial markers (COI and COII), the nuclear markers AMD and HB also supported the reciprocal monophyly of three or at least two of the lineages, respectively. The use of both ABGD algorithm and coalescence-based Bayesian analysis, using single or multi-locus strategies, reinforced the subdivision of the M. projectans complex into at least three clusters, with an exact correspondence to the three lineages previously defined using DNA Barcoding. These results allowed rejecting the hypothesis that the clades observed by Machado et al. [25] would be a result of selective scanning associated with three different and recent invasions of Wolbachia in M. projectans [28]. Nevertheless, it is still possible that such patterns are resultant of ancient invasions independently fixed in different populations, that were first associated with cytoplasmic incompatibility, and post-zygotic isolation, and later enhanced the emergence or pre-zygotic isolation through reinforcement [91-93]. In this case, Wolbachia could act as a pro-speciation factor, and such a longstanding infection could result in fixation also for nuclear alleles. In fact, there is evidence for Wolbachia infection in at least one species of the Zygothrica genus group, and this endosymbiont was shown to be more frequent in mycophagous than non-mycophagous Diptera [29]. Although at a first sight simple diagnostic amplification could be used to confirm the presence of Wolbachia in our samples, shortages related to the amplification success presented by some of the markers employed here would not allow a confident description of prevalence patterns with such a methodology. Thus, it remains to be further evaluated if Wolbachia could or not be involved in the speciation of the M. projectans complex. Even so, this uncertainty does not affect the conclusion about the presence of at least three species under the general morphotype previously ascribed to M. projectans.”

P5: “... M. projectans affinis 2 presents higher suitability for this region [Dominican Republic].” This is very difficult to see in the figure. Is there a way to easily quantify this that shows higher suitability relative to the other species?

This was now tested with an ANOVA test, as stated at lines (574-577): “mean logistic suitability of 0.67, against 0.57 and 0.50 presented for M. projectans affinis 1 and M. projectans affinis 1, respectively (F = 4.74, p < 0.05, in an ANOVA performed with suitability values presented by each species across 25 random points chosen around the area”.

P8: “...seems to have been hold despite ...” -> “ ...seems to have been maintained despite ...”

Corrected.

” ...allows us hypothesizing ...” -> “ ...allows us to hypothesize ...”

Corrected.

Reviewer 3:

Reviewers Comments to the Author

 I think the biological problem is interesting, but the sample size to answer the biological question is not significant. The sampled area is restricted to interior south Atlantic Forest and Pampa and COI mt sequences are predominant. I think the authors have data to elaborate hypotheses about diversification patterns for the group but not to discuss process.

This sampling size was only accomplished after several collections performed in different Brazilian municipalities. Specimens of the M. projectans complex were found in 45 localities, but sample sizes in some cases was reduced to a few individuals. Furthermore, only males were used due to shortcomings related to the effectiveness of the identification process (see above). When DNA was finally extracted, frequent amplification failures were faced, which commonly lead to DNA termination (see above). Thus, we really understand the reasoning of the reviewer, but we can ascertain that several different trials were performed to circumvent these biases. Our results were attained after an intense effort, and we think they provided several interesting clues about the diversity and the evolution of the complex, although it leaves some interesting open questions. 

Reviewer 4:

Reviewers Comments to the Author

This paper examines the phylogeography of the Mycodrosophila projectans species complex, a mycophagous lineage in the Zygothrica Genus Group. The authors have thoroughly analyzed a combination of mitochondrial and nuclear gene data to discover three cryptic species in this complex. Overally, I found this a very easy paper to read and review. The authors did a wonderful job explaining all of their analyses. I only have a couple minor comments about the dating analyses.

→ How did you select the outgroup taxa for this study? This isn’t really justified in the paper and the species selected seem like a random mix of taxa across the phylogeny, with a slight focus on the Hawaiian Drosophila-Scaptomyza lineage. Was this done to facilitate dating analyses? 

We are really grateful for your comments and suggestions. The selection of the outgroup taxa is now explained in lines (264-270): “additional sequences were downloaded from GenBank for D. melanogaster (Sophophora sub-genus), D. virilis (Siphlodora subgenus), D. ornatifrons (Drosophila subgenus), D. grimshawii and Scaptomyza (Hawaiian drosophilids) from the Drosophilinae subfamily, and P. variegata from Steganinae subfamily (see S2 Table). These species represent a set of different lineages within Drosophilidae phylogeny [78], and were employed as successive outgroups as well as to establish further calibration points within our phylogeny (see below).”

→ The Tamura and Survov dates are quite different from one another. Perhaps this might warrant some discussion? Obbard et al. (2012) do discuss a wide range of dates and it might be helpful to refer to this paper in your discussion.

The use of Tamura and Suvorov dates is now discussed, and the results of Obbard et al. (2012) are referred in the paper (see lines 611-625): “This divergence timing was established using a fossil calibration [79,80], and two contrasting molecular clock estimates provided by Tamura et al. [81] and Suvorov et al. [82] for the divergence between Sophophora and the remaining Drosophila. The first of these studies employed 176 nuclear genes to estimate a genomic mutation clock rate, using the time of formation of Kauai Island in Hawaii as a prior to the divergence of D. picticornis and other species of the planitibia group of Hawaiian drosophilids [97]. Suvorov et al. [82] employed 155 genome assemblies to generate a matrix of 2,791 genes, which was used to reconstruct a fossil calibrated phylogeny, based on at least five fossil datings. As biogegraphical calibrations can overestimate divergence timings, leading to underestimations of rates [98], whereas fossil calibrations can underestimate divergence timings, leading to overestimated rates [99], we preferred to use intermediate values to accommodate these biases. Interestingly, using these priors, we obtained a mean divergence time of 39.7 Mya [95% HPD Interval 24.5 – 55.4 Mya] which is quite similar to the results obtained by Obbard et al. [98] using a mutation rate establish for four-fold degenerate codons, which added further confidence to our results.”

---

## [Editor Report · Decision Letter 1]

26 Jan 2022

PONE-D-21-28403R1Unveiling the Mycodrosophila projectans (Diptera, Drosophilidae) species complex: insights into the evolution of three Neotropical cryptic and syntopic speciesPLOS ONE

Dear Dr. Robe,

Thank you for submitting your manuscript to PLOS ONE. After careful consideration, we feel that it has merit but does not fully meet PLOS ONE’s publication criteria as it currently stands. Therefore, we invite you to submit a revised version of the manuscript that addresses the points raised during the review process.

The authors have done a good job in responding to the peer reviewers' comments, but the authors did not follow the instructions listed in the initial decision letter to address all of the English language problems. I think the paper contains some interesting and verifiable results, but the Discussion needs to be shortened and all speculation and redescriptions of the Results section need to be eliminated. I have uploaded an edited copy of your MS with many specific changes and questions. Several sections and paragraphs in the Discussion section should be deleted. Overall, the writing style is wordy with much jargon. It reads like a thesis. Again, I suggest the MS be proofread by an English speaker before resubmission because I may not have found all of the grammar and wording problems.

We look forward to receiving your revised manuscript.

Kind regards,

William J. Etges

Academic Editor

PLOS ONE

---

## [Author Response · Author response to Decision Letter 1]

10 Mar 2022

Santa Maria, March 08th, 2022.

Dear Dr. William J. Etges,

Please find attached the MS entitled “Unveiling the Mycodrosophila projectans (Diptera, Drosophilidae) species complex: insights into the evolution of three Neotropical cryptic and syntopic species” which contains a revised version of the previous submission to PLOS ONE (PONE-D-21-28403). We thank you for the opportunity as well as for the constructive criticism, which allowed several improvements in our study.

We carefully followed the recommendations and provide below the explanation (in bold) on how we proceeded about each specific comment (in italics). Excerpts from the text are represented in yellow in this letter, and are highlighted in the tracked version of the MS.

We are confident the current version is much improved and hope it reaches the level of quality expected by PLOS ONE.

Sincerely,

L. J. Robe, on behalf of all authors

Editor`s comment:

“The authors have done a good job in responding to the peer reviewers' comments, but the authors did not follow the instructions listed in the initial decision letter to address all of the English language problems. I think the paper contains some interesting and verifiable results, but the Discussion needs to be shortened and all speculation and redescriptions of the Results section need to be eliminated. I have uploaded an edited copy of your MS with many specific changes and questions. Several sections and paragraphs in the Discussion section should be deleted. Overall, the writing style is wordy with much jargon. It reads like a thesis. Again, I suggest the MS be proofread by an English speaker before resubmission because I may not have found all of the grammar and wording problems.”

Thank you for your effort in thoroughly reviewing our manuscript. All the changes pointed out in the pdf were performed, and the MS was proofread by a specialized company (see certificate attached to the submission). We also withdrawn several speculations of the Discussion, as well as the redescriptions of the results. Thus, the last section was significantly shortened.

Concerning each commentary presented in the pdf:

Abstract – line 39: Look up the definition of niche.

As suggested, we withdrawn the reference to “biotic and abiotic niches”, so that the mentioned sequence now looks like: “Ecologically, sympatry and syntopy seem to be recurrent for these three cryptic species, which present widely overlapping niches, implying niche conservatism.” (lines 34-36)

Introduction – lines 69-71:

“This last sentence needs rewriting or deletion. There are many reasons beyond non-visual signals why cryptic species exist. Why is vision so important? All kinds of signaling systems are known. They could have diverged in allopatry and then converged later.”

We decided to rewrite the sentence, as follows: “The mechanisms responsible for this pattern are not well understood [13], but it has been suggested that sexual selection acting at the level of non-visual signals [14], recent divergence [15], chance fixation of alleles related to epistatic incompatibilities [16], and allopatric speciation associated with morphological stasis or convergent evolution [17] may help to explain the ubiquity of this phenomenon.” (lines 56-60)

Materials and Methods – line 139: State what you mean by “active period”.

The sentence was rewritten to: “In these localities, active searches for fruiting bodies of macroscopic fungi were performed in forest fragments during daylight.” (lines 116-118)

Results – line 357: What is “Bold system”? Define.

BOLD is defined in line 164 of the Materials and Methods: “All individuals were identified by the DNA barcoding in BOLD (Barcode of Life Data) System [51] using COI sequences.”

Results – title presented in lines 433-433: replace “What can we say about the differential patterns of genetic diversity and abiotic niches that distinguish the three species and what this suggests about their speciation?” to “Patterns of genetic diversity and abiotic niches distinguish the three species and suggest modes of speciation”.

The title was rewritten to: “Patterns of distribution of genetic diversity and abiotic niches help to disentangle evolutionary history and suggest modes of speciation” (lines 375-377)

Discussion – lines 569-591: These are a restatemnt of the results. Delete.

The paragraph was withdrawn, and only two sentences were left to mention the cryptic morphology: “The cryptic morphology among the three species was confirmed after rigorous analysis, and only a few subtle morphological differences were found for the three species in terms of aedeagus morphology and abdominal color patterns. However, none of these allowed the three species of the complex to be unambiguously distinguished” (lines 499-502). Otherwise, this aspect would not be mentioned in the discussion, and there would be a gap to the comprehension of the next paragraphs.

Discussion – lines 595-596, 600-601: Previous sentence deleted. Its irrelevant and speculative.

Both sentences were deleted, as well as all the section dealing with preferential mating. The new section now looks like: “In addition to this cryptic morphology, the M. projectans complex also detaches for recurrent sympatry and syntopy among its species, as observed here in three samples from the Atlantic Forest and the Pampa biome. The coexistence of cryptic species seems to be a common pattern in Mycodrosophila since Lacy [92] also reported syntopy for two cryptic species in the Nearctic (M. claytonae A and B). Within the Zygothrica genus group, the co-occurrence of cryptic species has also been reported for Neotropical species of the genera Zygothrica and Hirtodrosophila [28, 93]” (lines 502-508). This paragraph was further associated with the next one, continuing with: “Nevertheless, for the M. projectans complex, beyond co-occurrence, patterns of biotic and abiotic niches were also quite similar, and species used the same genera of fungi as breeding sites. Despite differences among species concerning the amplitude of potential distribution, neither pairwise comparison between abiotic niches recovered through ENM strategies rejected the null hypothesis of niche equivalency. These results support the hypothesis of niche conservatism for the M. projectans complex [29], suggesting that it may be a common pattern in the Zygothrica genus group. Nevertheless, further ecological studies are necessary to clarify in which way these species use their resources and how competition affects their co-occurrence” (lines 509-517).

Lines 624-61, and 632-637 were also deleted, and the two next paragraphs were shortened and associated: “Patterns of morphological similarity and niche equivalency seem to have been maintained despite ancient divergence. In this sense, although there were incongruences among gene trees regarding the topology of the relationships within the M. projectans complex, the multi-locus strategy used to reconstruct the species tree under a relaxed molecular clock strategy allowed us to date the two cladogenesis events to approximately 17–15 Mya ago. These estimates suggest that the diversification of the M. projectans complex occurred in the Neogene period during the Miocene [94]. This contrasts with the pattern commonly found for Drosophilidae, in which cryptic species encompass recently diverged lineages [93, 95, 96] that did not have enough time to accumulate diagnosable features. Thus, this study provides an interesting example of morphological conservation, despite ancient divergence. Furthermore, this ancient divergence, in view of the similarity of niches evidenced here for the three lineages, allows us to hypothesize that divergence of the species occurred in allopatry or parapatry, with current records of sympatry and syntopy reflecting secondary contact. This hypothesis of allo- or parapatric distribution in more ancient times was also supported by EBSP analyses, which showed contrasting patterns of demographic fluctuations for all species. In fact, whereas M. projectans affinis 1 did not present significant signals of population expansion, such signals were supported for M. projectans affinis 2 and M. projectans affinis 3 as occurring at different moments of the Middle Pleistocene [94], ca. 200 and 400 Kya, respectively. Such differences would certainly not be expected if species were sympatric, especially in the face of straightforward evidence of niche equivalence” (lines 518-537)

Discussion, line 655-656: Replace “these three species seem to have expanded to the sampled region quite recently” to “these three species seem to have expanded quite recently”.

In this sentence, we preferred to adopt an intermediate version stating that ”these three species seem to have expanded to Southern Brazil quite recently” (lines 538-539), since our modelling results clearly suggest an expansion to this specific area.

---

## [Editor Report · Decision Letter 2]

23 Mar 2022

PONE-D-21-28403R2Unveiling the Mycodrosophila projectans (Diptera, Drosophilidae) species complex: insights into the evolution of three Neotropical cryptic and syntopic speciesPLOS ONE

Dear Dr. Robe,

Thank you for submitting your manuscript to PLOS ONE. After careful consideration, we that it does not fully meet PLOS ONE’s publication criteria as it currently stands. Therefore, we invite you to submit a revised version of the manuscript that addresses the points raised by the AE.

Thank you for your revised submission, but on careful rereading, the authors have not incorporated all of the suggested wording changes. Also, no marked up MS in track changes was included making it extremely difficult to evaluate the edits the authors have made. A MS with yellow highlighted sections of revised sentences was included, but it does not show all the line by line wording and phrasing edits plus sections deleted that are required. Further, the editing service used has apparently used "British English" during the editing process - PLoS journals are published in the USA.

We look forward to receiving your revised manuscript.

Kind regards,

William J. Etges

Academic Editor

PLOS ONE
---

## [Author Response · Author response to Decision Letter 2]

21 Apr 2022

Santa Maria, April 21th, 2022.

Dear Dr. William J. Etges,

Please find attached the MS entitled “Unveiling the Mycodrosophila projectans (Diptera, Drosophilidae) species complex: insights into the evolution of three Neotropical cryptic and syntopic species” which contains a revised version of the previous submission to PLOS ONE (PONE-D-21-28403). We thank you for the opportunity as well as for the constructive criticism, which allowed several improvements in our study.

We carefully followed the recommendations and provide below the explanation (in bold) on how we proceeded about each specific comment (in italics). Excerpts from the text are represented in yellow in this letter and are highlighted in blue in the tracked version of the MS. Words or sentences that have been pointed to present grammar problems were edited according to suggestions provided in the pdf file and are highlighted in blue in the tracked version of the MS. When there is an overlap of both circumstances, the modified words or sentences are presented in green in the MS. Finally, specific suggestions that could not be accepted are presented in red, and justification for this is provided bellow.

We also detach that the manuscript was proofread by a specialized company (see certificate attached to the submission). Although the format style hired from this company was British, suggestions which did not follow USA English were not implemented. 

We are confident the current version is much improved and hope it reaches the level of quality expected by PLOS ONE.

Sincerely,

L. J. Robe, on behalf of all authors

Editor`s comment:

“The authors have done a good job in responding to the peer reviewers' comments, but the authors did not follow the instructions listed in the initial decision letter to address all of the English language problems. I think the paper contains some interesting and verifiable results, but the Discussion needs to be shortened and all speculation and redescriptions of the Results section need to be eliminated. I have uploaded an edited copy of your MS with many specific changes and questions. Several sections and paragraphs in the Discussion section should be deleted. Overall, the writing style is wordy with much jargon. It reads like a thesis. Again, I suggest the MS be proofread by an English speaker before resubmission because I may not have found all of the grammar and wording problems.”

Thank you for your effort in thoroughly reviewing our manuscript. Most of the changes pointed out in the pdf were performed, and the MS was proofread by a specialized company (see certificate attached to the submission). We have also withdrawn several speculations of the Discussion, as well as the redescriptions of the results. Thus, the last section was significantly shortened.

Concerning each commentary presented in the pdf:

Abstract – line 39: Look up the definition of niche.

As suggested, we withdrawn the reference to “biotic and abiotic niches”, so that the mentioned sequence now looks like: “Ecologically, sympatry and syntopy seem to be recurrent for these three cryptic species, which present widely overlapping niches, implying niche conservatism.” (lines 34-36)

Introduction – lines 69-71:

“This last sentence needs rewriting or deletion. There are many reasons beyond non-visual signals why cryptic species exist. Why is vision so important? All kinds of signaling systems are known. They could have diverged in allopatry and then converged later.”

We decided to rewrite the sentence, as follows: “The mechanisms responsible for this pattern are not well understood [13], but it has been suggested that sexual selection acting at the level of non-visual signals [14], recent divergence [15], chance fixation of alleles related to epistatic incompatibilities [16], and allopatric speciation associated with morphological stasis or convergent evolution [17] may help to explain the ubiquity of this phenomenon.” (lines 56-60)

Materials and Methods – line 139: State what you mean by “active period”.

The sentence was rewritten to: “In these localities, active searches for fruiting bodies of macroscopic fungi were performed in forest fragments during daylight.” (lines 116-118)

Results – line 357: What is “Bold system”? Define.

BOLD is defined in line 164 of the Materials and Methods: “All individuals were identified by the DNA barcoding in BOLD (Barcode of Life Data) System [51] using COI sequences.”

Results – title presented in lines 433-433: replace “What can we say about the differential patterns of genetic diversity and abiotic niches that distinguish the three species and what this suggests about their speciation?” to “Patterns of genetic diversity and abiotic niches distinguish the three species and suggest modes of speciation”.

The title was rewritten to: “Patterns of distribution of genetic diversity and abiotic niches help to disentangle evolutionary history and suggest modes of speciation” (lines 376-378)

Discussion – lines 569-591: These are a restatement of the results. Delete.

The paragraph was withdrawn, and only two sentences were left to mention the cryptic morphology: “The cryptic morphology among the three species was confirmed after rigorous analysis, and only a few subtle morphological differences were found for the three species in terms of aedeagus morphology and abdominal color patterns. However, none of these allowed the three species of the complex to be unambiguously distinguished” (lines 499-502). Otherwise, this aspect would not be mentioned in the discussion, and there would be a gap to the comprehension of the next paragraphs.

Discussion – lines 595-596, 600-601: Previous sentence deleted. It’s irrelevant and speculative.

Both sentences were deleted, as well as all the section dealing with preferential mating. The new section now looks like: “In addition to this cryptic morphology, the M. projectans complex also detaches for recurrent sympatry and syntopy among its species, as observed here in three samples from the Atlantic Forest and the Pampa biome. The coexistence of cryptic species seems to be a common pattern in Mycodrosophila since Lacy [92] also reported syntopy for two cryptic species in the Nearctic (M. claytonae A and B). Within the Zygothrica genus group, the co-occurrence of cryptic species has also been reported for Neotropical species of the genera Zygothrica and Hirtodrosophila [28, 93]” (lines 502-508). This paragraph was further associated with the next one, continuing with: “Nevertheless, for the M. projectans complex, beyond co-occurrence, patterns of biotic and abiotic niches were also quite similar, and species used the same genera of fungi as breeding sites. Despite differences among species concerning the amplitude of potential distribution, neither pairwise comparison between abiotic niches recovered through ENM strategies rejected the null hypothesis of niche equivalency. These results support the hypothesis of niche conservatism for the M. projectans complex [29], suggesting that it may be a common pattern in the Zygothrica genus group. Nevertheless, further ecological studies are necessary to clarify in which way these species use their resources and how competition affects their co-occurrence” (lines 509-516).

Lines 624-61, and 632-637 were also deleted, and the two next paragraphs were shortened and associated: “Patterns of morphological similarity and niche equivalency seem to have been maintained despite ancient divergence. In this sense, although there were incongruences among gene trees regarding the topology of the relationships within the M. projectans complex, the multi-locus strategy used to reconstruct the species tree under a relaxed molecular clock strategy allowed us to date the two cladogenesis events to approximately 17–15 Mya ago. These estimates suggest that the diversification of the M. projectans complex occurred in the Neogene period during the Miocene [94]. This contrasts with the pattern commonly found for Drosophilidae, in which cryptic species encompass recently diverged lineages [93, 95, 96] that did not have enough time to accumulate diagnosable features. Thus, this study provides an interesting example of morphological conservation, despite ancient divergence. Furthermore, this ancient divergence, in view of the similarity of niches evidenced here for the three lineages, allows us to hypothesize that divergence of the species occurred in allopatry or parapatry, with current records of sympatry and syntopy reflecting secondary contact. This hypothesis of allo- or parapatric distribution in more ancient times was also supported by EBSP analyses, which showed contrasting patterns of demographic fluctuations for all species. In fact, whereas M. projectans affinis 1 did not present significant signals of population expansion, such signals were supported for M. projectans affinis 2 and M. projectans affinis 3 as occurring at different moments of the Middle Pleistocene [94], ca. 200 and 400 Kya, respectively. Such differences would certainly not be expected if species were sympatric, especially in the face of straightforward evidence of niche equivalence” (lines 517-536)

Discussion, line 655-656: Replace “these three species seem to have expanded to the sampled region quite recently” to “these three species seem to have expanded quite recently”.

In this sentence, we preferred to adopt an intermediate version stating that “these three species seem to have expanded to Southern Brazil quite recently” (lines 537-538), since our modelling results clearly suggest an expansion to this specific area.

Concerning the English grammar corrections suggested on the pdf and the spelling review, most suggestions were accepted, and are detached in yellow or green in the text with tracked changes. There were only some specific cases in which the suggestion was not followed, and these cases are marked in red in the text with tracked changes. Bellow we detail each particular situation (see that lines reference refer to the reviewed version):

Abstract:

→ Line 25: Replace “The Zygothrica genus group” by “The Zygothrica species group” � This replacement would not be correct, since the target taxon is really a genus group, as proposed by Grimaldi (1990).

→ Line 25: “to be a speciose group” by “to be speciose” � Ok.

→ Line 26: “have revealed” by “have been” � Ok.

→ Line 28: “The aim of this study is…” by “The aim of this study was” � Ok.

→ Line 32: “… described for…” by “described as” � Ok.

→ Lines 33-34: “...In fact, only some subtle morphological differences could be encountered for the three species when aedeagus patterns are evaluated in association with abdominal color patterns.” was replaced by “Only a few subtle morphological differences were found for the three species in terms of aedeagus morphology and abdominal color patterns”, as requested.

→ Lines 33-34: “… seems to be a recurrent pattern…” was replaced by “seem to be recurrent”, as suggested.

 → Lines 35-36: “… which present widely overlapped biotic and abiotic niches, in an intrinsic pattern of niche conservatism.” was replaced by “which present widely overlapping niches, implying niche conservatism.”

→ Lines 36-37: Replace “This straightforward morphological and ecological similarity occurs even though cladogenesis” by “This morphological and ecological similarity has persisted though cladogenesis” � Ok.

→ Lines 37: Replace “seems to have taken place during Miocene” by “dates back to the Miocene” � Ok.

→ Line 39: Replace “added to some contrasting patterns” by “in addition to contrasting patterns” � Ok.

→ Lines 39-40: Replace “allowed hypothesizing allopatric or parapatric diversification” to “allowed us to hypothesize patterns of allopatric or parapatric diversification” � Ok.

→ Lines 40-41: Replace “with expansion to the sampled region leading to secondary contact” by “with expansion within their current geographic ranges” � Because our samples did not encompass the whole geographic ranges of each species, we preferred to change this sentence to “with secondary contact in Southern Brazil”.

Introduction:

→ Line 49: “...the world biodiversity…” was changed to “...the world’s biodiversity...”, as suggested.

→ Line 51: “... processes responsible for the origin of all this diversity have been largely discussed…” was changed to … “processes responsible for the origin of this diversity have been previously discussed...”, as suggested.

→ Line 52-53: “The molecular taxonomy…” was changed to “Molecular taxonomy…”, as requested. 

→ Line 57: Replace “largely misunderstood” by “not well understood” � Ok.

→ Line 57: Replace “Nevertheless” by “but” � Ok.

→ Line 61: Replace “Drosophilidae is a speciose family of insects, presenting …” by “The family Drosophilidae includes…” � Ok. 

→ Line 62: Replace “Flies of this family are …” by “Different species are…” � Ok. 

→ Line 64: Replace “… occurs due…” by “… is largely due…” � Ok.

→ Lines 64-65: Replace “… which feed mainly on bacteria and yeast involved in the fermentaion of each of these resources or compounds thereby made available…” by “… that feed mainly on bacteria and yeasts involved in fermentation and on fermentation by-products…” � Ok. That really sounds better. Thank you.

→ Lines 65-66: Replace “… many taxa of the family…” by “… many species…” � Ok.

→ Line 67: Replace “This is the case of…” by “This is the case for…” � Ok.

→ Line 68: Replace “The Zygothrica genus group” by “The Zygothrica species group” � This would not be correct, since the target taxon is really a genus group, as proposed by Grimaldi (1990).

→ Line 74: Replace “… reported to…” by “reported in…” � Ok.

→ Line 76: Replace “… registered for…” by “described in…” � Ok.

→ Line 76: Replace “… practically doubled…” by “nearly doubled…” � Ok.

→ Line 77: Replace “… thanks to results obtained through…” by “…due to…” � Ok.

→ Line 77: Replace “… effort…” by “… efforts…” � Ok.

→ Line 78: Replace “In fact, this last approach also suggested…” by “The latter approach suggested…” � Ok.

→ Line 86: “Nevertheless,…” was changed to “However,...” as requested. 

→ Line 87: Replace “… paralogous…” by “… paralogues…” � Ok.

→ Line 88: Replace “… or even due to “hitchhik…” by “… hitchhiking…” � Ok.

→ Line 89: Replace “In fact, whereas the first phenomenon” by “The presence of numts…” � Ok.

→ Line 92: Delete “Furthermore, …” � Ok.

→ Lines 94-95: “… and as several species of insects can occur in a single fungus [29], mushrooms can be considered a hotspot for horizontal transfer of this endoparasite” was rewritten to “...therefore, mushrooms may be hotspots for horizontal transfer of this endoparasite, where multiple drosophilid species co-exist [32]”, as requested.

→ Lines 94-97: “In several cases, only the addition of data from nuclear gene or morphological markers could allow to assess the interference of this phenomena in the evolutionary history of the characterized mitochondrial genes” was rewritten to “Including nuclear gene sequences or morphological markers, along with mitochondrial genes, may allow insight into these evolutionary relationships (lines 94-97)”, as requested.

→ Line 98: “In this context, the aim of the present study is to confirm the subdivision of the M. projectans complex into at least three species…” was changed to “The aim of the present study was to confirm the subdivision of the M. projectans complex”, as suggested.

→ Line 99: “...related to their” was changed to “related to its”, as suggested.

→ Line 100: Replace “In this sense, we increased the sampling…” by “We increased sampling” � Ok.

→ Line 103: Replace “These strategies…” by “These analyses…” � Ok.

→ Lines 105-106: “Even so, the straightforward morphological and ecological similarity of the three species is astonishing, especially in the face of divergence dated to the Miocene.” was changed to “The morphological and ecological similarity of the three species was notable, especially in the face of divergence dating back to the Miocene”, as suggested.

Materials and methods

→ Line 110: Replace “… performed…” by “… made…” � Ok.

→ Line 111: Replace “… Km2…” by “… km2…” � Ok.

→ Line 112: Delete “… of South America area” � Ok.

→ Line 113: Replace “… covers around…” by “… includes ca.…” � Ok.

→ Line 113: Replace “… along Brazil’s coastal area…” by “… along coastal Brazil…” � Ok.

→ Lines 114-115: Replace “… Rio Grande do Sul state…” by “… state of Rio Grande do Sul…” � Ok.

→ Line 116: Replace “…tree formations…” by “… forests…” � Ok.

→ Line 118: Replace “…adult specimens…” by “… adults…” � Ok.

→ Line 118: Replace “…with the use of an entomological aspirator…” by “… with an aspirator…” � Ok.

→ Line 119: Replace “…the resources were carried to the laboratory…” by “… fungi were returned to the laboratory…” � Ok. 

→ Line 119: Replace “…until adults emergence…” by “… until adults emerged…” � Ok.

→ Line 121: Replace “Just males were employed in further analyses…” by “Only males were used…” � Ok.

→ Line 121-122: “…because morphological differences in Drosophilidae are frequently described or even found through inspection of the internal genital structures of males…” was replaced by “because morphological differences at the aedeagus level provide more accurate species identification”, which is a little different from the Editor suggestion but still describes accurately the information.

→ Line 128: Delete “The…” at the beginning of the sentence � Ok.

→ Line 128: Replace “… each male specimen…” by “… each adult male…” � Ok.

→ Line 130: Replace “COI gene was…” by “COI genes were…” � Ok.

→ Line 142: Add “the” before “Staden package” � Ok.

→ Line 143: Replace “… editions were performed for…” by “… editing…” � Ok.

→ Line 148: Delete “Additionally, for these datasets, …” � Ok.

→ Lines 154-155: Replace “… to suggest M. projectans as…” by “… suggesting M. projectans was…” � Ok.

→ Line 156: Replace “… in the present work…” by “… here…” � Ok.

→ Line 156: Replace “… In the morphological approach, evaluations were performed in two ways: first, by external morphology, in which males were photographed with a stereoscopic microscope…” by “First, males were photographed with a stereoscopic microscope …” by “… here…” � Ok.

→ Line 157: Replace “… right wings were…” by “… each right wing was…” � Ok.

→ Line 158: Replace “… ; second, by…” by “… Second, using…” � Ok.

→ Lines 158-159: Replace “… to search for differences in the male aedeagus structures based on Wheeler & Takada [22] descriptions…” by “… we assessed differences in male aedeagus morphology based on Wheeler & Takada…” � Ok.

→ Lines 159-60: Replace “For this task, male terminalia was…” by “… male termnalia were…” � Ok.

→ Line 160: Replace “… potassium hydroxide (KOH) 10% and ethanol 70%…” by “… 10% potassium hydroxide (KOH) and 70% ethanol…” � Ok.

→ Line 161: Replace “… further disjointed…” by “… dissected…” � Ok.

→ Line 161: Transfer “Leica” to the beginning of the name of the microscope � Ok.

→ Line 162: Delete “To this task, …” � Ok.

→ Line 162: Add “… a …” before 20x � Ok.

→ Line 162: Replace “… under excitation wavelengths of…” by “… with a…” � Ok.

→ Line 163: Replace “… at…” by “… to…” � Ok. 

→ Line 163: Delete “The evaluated morphological markers are important to Drosophilidae species delimitation and recognition and usually provide information to differentiate species [20, 22, 48-52].” at the end of the paragraph � Ok.

→ Line 164: Replace “Under the molecular approach, all individuals were first…” by “… all individuals were…” � Ok. 

→ Line 167: Replace “… the reciprocal monophyly…” by “… reciprocal monophyly…” � Ok. 

→ Lines 173-174: Replace the “[…]” by “, …,” for the text explaining precedence of the sequence MycoNova � Ok. 

→ Line 180: Add “the” before “Stacey package” � Ok.

→ Line 186: Add “the” before “Splits package” � Ok.

→ Line 192: Replace “200.000.000” and “10.000” by “200,000,000” and “10,000” respectively � Ok. 

→ Line 194: Replace “… specimens assignments…” by “… species assignments…” � Ok. 

→ Line 202: Replace “… intra…” by “… intra-…” � Ok. 

→ Line 208: Delete “For the morphometric analysis, …” at the beginning of the paragraph � Ok. 

→ Line 208: Replace “… male specimens…” by “…males…” � Ok. 

→ Line 208: Replace “… were photographed in…” by “…were photographed with…” � Ok. 

→ Line 210: Delete “To obtain the coordinates, …” at the beginning of the sentence � Ok. 

→ Lines 210-211: Replace “… chosen with basis on wings venation…” by “…chosen based on wing venation…” � Ok. 

→ Line 212: Add “a” before “with matrix” � Ok. 

→ Lines 213-214: Replace “An analysis of variance…” by “Analysis of variance…” � Ok. 

→ Line 225: Replace “An analysis of molecular variance…” by “Analysis of molecular variance…” � Ok. 

→ Lines 213-214: Replace “…performed in the same software…” by “…performed with the same software…” � Ok. 

→ Line 236: Correct “coalescent” � Ok.

 → Line 243: Replace “… within Drosophilidae phylogeny…” by “…within the Drosophilidae…” � Ok. 

→ Lines 247-250: Replace the “[…]” by “, …,” for the text explaining the priors and include these within the main text with expressions like “using a prior with a mean of …” or “with a prior with a mean of…” � Ok. 

→ Line 252: Replace “… ploidy level…” by “… ploidy levels…” � Ok. 

→ Line 252: Replace “… presented by…” by “… for…” � “… presented by” was replaced by “of”. 

→ Line 254: Replace “… burning…” by “… burn-in…” � Ok. 

→ Line 254: Delete “After, …” at the beginning of the sentence � Ok. 

→ Line 254: Replace “… the run was…” by “… the run was then…” � Ok. 

→ Line 264: Replace “10.000” by “10,000” � Ok. 

→ Line 264: Replace “… burn in the first…” by “burning the first” � Ok. 

→ Line 276: Replace “As the cryptic diversity contained within the M. projectans complex was previously neglected in the literature” by “As cryptic diversity in the M. projectans species complex was previously neglected…” � Ok. 

→ Line 277: Replace “… recorded by…” by “… recorded in…” � Ok. 

→ Line 277: Delete “… in this analysis” at the end of the sentence � Ok. 

→ Line 277: Replace “… presence point…” by “… sampling point…” � Ok. 

→ Line 280: Add “the” before “dismo package” � Ok. 

→ Line 288: Replace “… Schoener’s (D) and Hellinger’s (I) …” by “… Schoener’s D and Hellinger’s I …” � Ok. 

→ Line 289: Replace “… are…” by “… were…” � Ok. 

Results

→ Line 295: Delete “In the present study…” in the beginning of the section � Ok. 

→ Line 296: Replace “… number…” by “… numbers…” � Ok. 

→ Line 299: Replace “No sequence presented…” by “No sequences contained…” � Ok. 

→ Lines 301-302: Replace “… presented low K2P distance” by “… showed low K2P distances” � Ok. 

→ Notes of Table 1: Replace “heterozigosity” by “… heterozygosity” � Ok. 

→ Line 305: Replace “How many species are there under the general M. projectans morphotype” by “How many M. projectans species are there” � Ok. 

→ Line 307: Replace “… the 172 COI sequences” by “… of the 172 COI sequences” � Ok. 

→ Line 307: Replace “:” by “,” � Ok. 

→ Line 309: Replace “[ ]” by “,” � Ok. 

→ Line 309: Replace “The total set of specimens…” by “All individuals” � Ok. 

→ Line 311: Replace “;” by “,” � Ok. 

→ Line 312: Replace “Moreover, to the exception of…” by “With the exception of…” � We preferred to adopt a more concise form “Except for”

→ Line 313: Replace “… the three species revealed…” by “… the three species were…” � Ok. 

→ Line 324: Replace “… for the COI marker…” by “… for COI…” � We transferred the name of the marker, which is now mentioned regarding “maximum COI intraspecific distances”. 

→ Line 324: Replace “… presented…” by “… showed…” � Ok. 

→ Line 325: Delete “On the other hand…” � Ok. 

→ Line 329: Replace “… presented…” by “… showed…” � Ok. 

→ Line 332: Replace “… is…” by “… was…” � Ok. 

→ Line 334: Replace “Although at a first sight specimens” by “… Individuals…” � Ok. 

→ Line 334: Replace “… are…” by “… were…” � Ok. 

→ Line 334: Add “… as…” before “… we could identify…” � We preferred to add “but” instead of “as”, since the sentence presents an opposing argument. 

→ Line 335: Replace “… two patterns shared…” by “… Two patterns were shared…” � Ok. 

→ Line 337: Replace “… are…” by “… were…” � Ok. 

→ Lines 337-339: Replace the “( )” by “,” to delimit the sentences “hereafter called abdominal pattern x” � Ok.

→ Line 341: Replace “… with exceptions to this pattern found in two individuals attributed to this species…” by “There were two exceptions to this pattern found in this species…” � Ok. 

→ Line 344: Replace “… confirms a close resemblance…” by “… confirmed the close resemblance…” � Ok. 

→ Line 346: Delete “The” before “wing” at the beginning of the sentence � Ok. 

→ Line 346: Replace “… tested…” by “… used…” � Ok. 

→ Line 346: Replace “… 57 specimens…” by “… 57 individuals…” � Ok. 

→ Line 350: Delete “as” before “a wide overlap” � Ok. 

→ Lines 351-352: Replace “… wings did not provide adequate resolution…” by “… wing morphology did not provide enough resolution…” � Ok. 

→ Line 353: Replace “At the male genital levels…” by “At the male genitalia levels…” � Ok. 

→ Line 357: Replace “… shared…” by “… was shared…” � Ok. 

→ Line 360: Replace “… with aedeagus longer…” by “… with the aedeagus longer…” � We preferred to replace the previous expression by “with a longer aedeagus”, because it fits better with the next reasoning.

→ Line 367: “… even the mitochondrial markers…” was replaced by “… even both mitochondrial markers…”, because the suggestion would change the meaning of the sentence.

 → Line 368: Replace “… as sister species…” by “… as a sister species” � We think this suggestion would not be adequate within the arrangement of the ideas.

→ Line 374: Replace “… whereas the next cladogenesis occurred shortly after, around 15 Mya.” by “… and the next split occurred ca. 15 Mya” � Ok.

→ Lines 376-378: The subtitle “What can we say about the differential patterns of genetic diversity and abiotic niche that distinguish the three species and what this suggests about their speciation” was replaced by “Patterns of distribution of genetic diversity and abiotic niches help to disentangle evolutionary history and suggest modes of speciation”, as suggested.

→ Line 380: Delete “as” before “for mitochondrial and nuclear markers, respectively” � Ok.

→ Lines 380-381: Replace the “( )” by “,” to delimit the sentence “for mitochondrial and nuclear markers, respectively” � Ok.

→ Line 382: Replace “presented” by “revealed” � Ok.

→ Line 382: Replace “attained” by “showed” � Ok.

→ Line 388: Replace “is” by “was” � Ok.

→ Line 390: Replace “is commonly very similar” by “was very similar” � Ok.

→ Lines 392-393: Replace the “… presented a star-like pattern…” by “showing star-like patterns” � Ok.

→ Line 394: Replace “are” by “were” � Ok.

→ Line 398: Replace “haplotype” by “haplotypes” � Ok.

→ Line 399: Replace “Although the differentiation…” by “Although differentiation…” � Ok.

→ Line 400: Replace “… seems to be maintained…” by “… seems to have been maintained…” � Ok.

→ Line 406: Replace “is” by “was” � Ok.

→ Line 409: Replace “… sampled in the border…” by “… sampling along the border…” � Ok.

→ Line 409: Add “The” before “Atlantic Forest” � Ok.

→ Line 410: Replace “… present…” by “… contained…” � Ok.

→ Line 411: Replace “… radiates…” by “… radiated…” � Ok.

→ Lines 417-418: Delete the information presented in parenthesis for the subdivision of M. projectans affinis 1 and affinis 3 � Ok.

→ Lines 419-420: Replace the “( )” by “,” to delimit the sentence “with two groups composed of individuals from the Atlantic Forest and another group exclusively composed of specimens from the Pampa biome” � Ok.

→ Line 419: Replace “… specimens…” by “… individuals…” � Ok.

→ Line 419: Add “The” before “Atlantic Forest” � Ok.

→ Line 421: Replace “… presented…” by “… revealed…” � Ok.

→ Line 424: Add “The results of…” before “the Extended Bayesian Skyline” � Ok.

→ Line 424: Replace “… seems to agree…” by “… agreed…” � Ok.

→ Line 425: Add “the” before “COI networks” � Ok.

→ Line 429: Replace “… ranges…” by “… ranged…” � Ok.

→ Line 434: Replace “… showed to be…” by “… are…” � Ok.

→ Line 434: Replace “… to an area…” by “… in an area…” � Ok.

→ Line 435: Replace “… Southern…” by “… southern…” � Ok.

→ Line 435: Delete “… their distribution can go much further, and specimens identified as…” � Ok.

→ Line 436: Replace “… have…” by “… have…” � Ok.

→ Line 437: Replace “… sampled area…” by “… areas sampled…” � Ok.

→ Line 437: Replace “… exploring fungi…” by “… associated with fungal…” � Ok.

→ Line 438: Replace “… the three species…” by “… all three species…” � Ok.

→ Line 442: Add “… in the sampled genera…” before Ganoderma � We inserted “in species of”.

→ Lines 444-445: Replace “. According to these results, although…” by “, suggesting…” � We preferred to separate the ideas in two sentences. Thus, we changed the subject of the second sentence, and the verb was conjugated to “suggested”.

→ Lines 445-446: Replace “… population overlaps in Southern Brazil are plausible not only for the present but also for different moments of the past…” by “… with present overlapping ranges in Southern Brazil and possibly at different times in past…” � We replaced previous sentence by “with present and past overlapping ranges”.

→ Line 447: Add “… the…” before Pleistocene � Ok.

→ Line 452: Delete “Nevertheless…” in the beginning of the sentence � Ok.

→ Line 455: Replace “Anyway…” by “Thus…” � We replaced “Anyway…” by “However…”, because these are contrasting results.

Discussion

→ Line 459: Delete “DNA Barcoding proved here to be a great tool for the identification and discovery of species and for understanding diversity in the Zygothrica genus group, as previously shown by Machado et al. [25]. Moreover…” at the beginning of the paragraph � Ok.

→ Line 459: Replace “… presents…” by “… revealed…” � Ok.

→ Line 459: Replace “… suggested…” by “… hypothesized…” � Ok.

→ Line 460: Replace “… a high incidence…” by “… a number…” � We used “several” instead of “a number of” to favor a concise language.

→ Line 460: Delete “Nevertheless…” at the beginning of the sentence � Ok.

→ Line 463: Replace “…hitchhike with the selective pressure acting on…” by “..hitchhiking due to” � Ok.

→ Line 464: Replace “… both artifacts…” by “both factors” � Ok.

→ Line 464: Replace “… proposition of the three species of the M. projectans complex…” by “… proposed inclusion of three species in the M. projectans complex” � Ok.

→ Line 465: Replace “… discarded…” by “… discounted…” � Ok.

→ Lines 465-466: Replace “… by the inclusion of evidence provided by this study…” by “… by the evidence provided in the current study…” � Ok.

→ Line 467: Delete “At first, the…” at the beginning of the paragraph � Ok.

→ Line 467: Replace “… diminishes…” by “… were diminished…” � Ok.

→ Line 472: Replace “… in our dataset…” by “… here…” � Ok.

→ Line 473: Replace “… it was possible to show that reciprocal monophyly and…” by “… we showed that reciprocal monophyly was consistent with…” � Ok.

→ Line 477: Replace “… occurs…” by “… occurred…” � Ok.

→ Line 478: Replace “In this study, besides…” by “In addition to…” � Ok.

→ Line 478: Delete “… by both mitochondrial markers…” and quote only “COI and COII” � Ok.

→ Line 480: Insert “the” before “ABGD” � Ok.

→ Line 483: Replace “… allowed rejecting…” by “… allowed us to reject…” � Ok.

→ Line 493: Delete “… at a first sight…” after “Although” � Ok.

→ Lines 494-495: “… shortages related to the amplification success presented by some of the markers employed here would not allow a confident description of prevalence patters with such a metodology” was replaced by “shortages related to the amplification success of some of our markers showed that such estimates of infection rates would not be precise”.

→ Line 496: Replace “… if…” by “… whether…” � Ok.

→ Line 496: Replace “… could or not be…” by “… was…” � Ok.

→ Line 496: Replace “… speciation…” by “… diversification…” � Ok.

→ Delete fourth and fifth paragraphs of the discussion � Ok. Please see that ideas previously presented on the sixth paragraph now appear on the third paragraph of the discussion.

→ Line 504: Replace “… sampling points located in…” by “… samples from…” � Ok.

→ Line 505: Delete “However, as sampling in several areas was not extensive, both sympatry and sintopy can be even more ubiquitous than shown” � Ok.

→ Line 509: Delete “This may suggest that species recognition and preferential mating may be acting at levels other than visual, as previously observed in Drosophila silvestris” � Ok.

→ Line 510: Replace “… biotic and abiotic niche…” by “… biotic and abiotic niches…” � Ok.

→ Line 510: Replace “… fungus…” by “… fungi…” � Ok.

→ Line 511: Delete “Thus,…” before despite � Ok.

→ Line 514: Delete “…, in agreement with Robe et al.” after referring to the hypothesis of niche conservatism for the M. projectans complex� Ok.

→ Line 516: Replace “… in which way the…” by “… how these…” � Ok.

→ Line 517: Delete “Interestingly,…” at the beginning of the paragraph � Ok.

→ Line 516: Replace “… the pattern…” by “Patterns…” � Ok.

→ Line 517: Replace “… divergence…” by “… divergences…” � Ok.

→ Line 517: Delete “This divergence timing was established using a fossil calibration [79,80], and two contrasting molecular clock estimates provided by Tamura et al. [81] and Suvorov et al. [82] for the divergence between Sophophora and the remaining Drosophila. The first of these studies employed 176 nuclear genes to estimate a genomic mutation clock rate, using the time of formation of Kauai Island in Hawaii as a prior to the divergence of D. picticornis and other species of the planitibia group of Hawaiian drosophilids [97]. Suvorov et al. [82] employed 155 genome assemblies to generate a matrix of 2,791 genes, which was used to reconstruct a fossil calibrated phylogeny, based on at least five fossil datings” � Ok.

→ Line 517: Replace “… underestimation…” by “… underestimations…” � The entire sentence was withdrawn in this reviewed version.

→ Line 522: Replace “According to our estimates,…” by “Our estimates indicated…” � Ok.

→ Line 526: Delete “Such a result may be explained by the presence of nonvisual mating signals and/or by the action stabilizing selection promoting morphological stasis [103].” Before Furthermore � Ok.

→ Line 529: Replace “… alo…” by “… -alo…” � Ok.

→ Line 530: Replace “… is…” by “… was…” � Ok.

→ Line 531: Replace “… for all the evaluated species” by “… for all species…” � Ok.

→ Line 537: Replace “In agreement with this…” by “Further…” � Ok.

→ Line 537: Replace “… the…” by “… these…” � Ok.

→ Line 538: Replace “… as evidenced…” by “… as shown…” � Ok.

→ Line 538: Replace “The high influence…” by “The influence…” � Ok.

→ Line 539: Replace “… supports…” by “… supported…” � Ok.

→ Line 540: Replace “… of these three species…” by “… of these species…” � Ok.

→ Line 541: Transfer “… in the Southern Hemisphere…” to the end of this sentence � Ok.

→ Line 549: Replace “… presenting evidence…” by “… suggesting…” � Ok.

→ Line 550: Replace “… to the…” by “… in…” � Ok.

→ Line 550: Replace “… further samplings are…” by “… further sampling is…” � Ok.

→ Line 553: Delete “Interestingly, …” at the beginning of the sentence � Ok.

→ Line 553: Delete “… in the sampled region …” after diversification � Ok.

→ Line 553: Delete “… (specially M. projectans affines 3) …” after complex � Ok.

→ Line 553: Replace “… show…” by “… showed…” � Ok.

→ Line 554: Replace “Southernmost” and “biomes” by “southernmost” and “biomes” � Ok.

→ Line 554: Add “the” before “Pampa” � Ok.

→ Line 555: Delete “This suggests the existence of intrinsic adaptations related to each of these habitats, which, to our knowledge, were not previously found for other Drosophilidae species. …” before “The levels…” Ok.

→ Line 556: Replace “… for the three species of the…” by “… for the…” � Ok.

→ Line 556: Replace “… are…” by “… were…” � Ok.

→ Line 559: Replace “… presented…” by “… showed…” � Ok.

→ Line 559: Delete “Although, these high diversity indices could suggest co-amplification of numts, tests performed to identify this artifact did not detect such an effect…” before “This high diversity…” Ok.

→ Line 559: Replace “… presented…” by “… showed…” � Ok.

→ Line 562: Delete “… in the haplotype networks of M. projectans affinis 2, it is possible to notice …” after “In this sense, …” and add this idea to the end of the sentence (lines 564-565) Ok.

→ Line 563: Replace “… are…” by “… were…” � Ok.

→ Line 563: Replace “… closer in relation to…” by “… more similar in…” � Ok.

→ Line 565: Replace “This suggests a metapopulation dynamics…” by “This suggests a possible metapopulation dynamics…” � Ok.

→ Line 565: Delete “…, which was already” after “dynamics” � Ok.

→ Line 566: Replace “… in virtue of…” by “… due to…” � We replaced the stretch by “in response to”.

Conclusion

→ Line 574: Replace “… exclude interferences such as numts…” by “… exclude numts…” � Ok.

→ Line 575: Replace “… was dated…” by “… dated…” � Ok.

→ Line 577: Replace “… the abiotic niche was identical…” by “… abiotic niches were equivalent…” � Ok.

→ Line 578: Replace “… generally presented…” by “… showed…” � Ok.

→ Line 579: Delete “In this sense,…” in the beginning of the last sentence � Ok.

→ Line 580: Replace “… this study certainly represents…” by “… this study represents…” � Ok.

→ Line 581: Replace “… comprehension of the spatio-temporal…” by “… understanding of potential spation-temporal…” � Ok.

---

## [Editor Report · Decision Letter 3]

4 May 2022

Unveiling the Mycodrosophila projectans (Diptera, Drosophilidae) species complex: insights into the evolution of three Neotropical cryptic and syntopic species

PONE-D-21-28403R3

Dear Dr. Robe,

We’re pleased to inform you that your manuscript has been judged scientifically suitable for publication and will be formally accepted for publication once it meets all outstanding technical requirements.

Kind regards,

William J. Etges

Academic Editor

PLOS ONE
---

## [Editor Report · Acceptance letter]

16 May 2022

PONE-D-21-28403R3 

Unveiling the Mycodrosophila projectans (Diptera, Drosophilidae) species complex: insights into the evolution of three Neotropical cryptic and syntopic species 

Dear Dr. Robe:

I'm pleased to inform you that your manuscript has been deemed suitable for publication in PLOS ONE. Congratulations! Your manuscript is now with our production department. 

Kind regards, 

on behalf of

Dr. William J. Etges 

Academic Editor

PLOS ONE